# Cumulative mitochondrial activity correlates with ototoxin susceptibility in zebrafish mechanosensory hair cells

**Sarah B Pickett[1,2], Eric D Thomas[1,2], Joy Y Sebe[1], Tor Linbo[1], Robert Esterberg[1,3], Dale W Hailey[1,3], David W Raible[1,2,3]***

[1]Department of Biological Structure, University of Washington, Seattle, United States; [2]Graduate Program in Neuroscience, University of Washington, Seattle, United States; [3]Virginia Merrill Bloedel Hearing Research Center, University of Washington, Seattle, United States

**Abstract** Mitochondria play a prominent role in mechanosensory hair cell damage and death. Although hair cells are thought to be energetically demanding cells, how mitochondria respond to these demands and how this might relate to cell death is largely unexplored. Using genetically encoded indicators, we found that mitochondrial calcium flux and oxidation are regulated by mechanotransduction and demonstrate that hair cell activity has both acute and long-term consequences on mitochondrial function. We tested whether variation in mitochondrial activity reflected differences in the vulnerability of hair cells to the toxic drug neomycin. We observed that susceptibility did not correspond to the acute level of mitochondrial activity but rather to the cumulative history of that activity.
DOI: https://doi.org/10.7554/eLife.38062.001

*For correspondence:
draible@uw.edu

Competing interests: The authors declare that no competing interests exist.

## Introduction

Neurons are some of the most energy demanding cells in the body (*Ames, 2000*). Because of this, they are particularly vulnerable to disruptions in mitochondrial and metabolic function. Despite the importance of proper mitochondrial function for all neuronal cell types, some subpopulations are more vulnerable than others, such as those affected by Parkinson's Disease, ALS, and other neurodegenerative diseases (*Saxena and Caroni, 2011*; *Jové et al., 2014*). This selective susceptibility of affected subpopulations has been attributed in part to their high physiological activity and altered response to oxidative stress. In addition to high activity levels, advanced age is also a major risk factor for neurodegenerative diseases, suggesting a connection between aging and selective cell death. Increased selective susceptibility over time could be attributed, at least in part, to the accumulation of oxidative damage caused by reactive oxygen species (ROS), as postulated by the free radical theory of aging (*Beckman and Ames, 1998*; *Finkel and Holbrook, 2000*; *Harman, 1956*). Mitochondria have been identified as a major source of ROS and studies have revealed that ROS production increases with age (*Beckman and Ames, 1998*; *Wallace, 2005*). If accumulation of mitochondrial oxidation and stress are related to cellular activity, highly active or older cells that have experienced more accumulated activity over time may be more vulnerable to acute damage.

Selective susceptibility is not limited to the central nervous system. Peripheral sensory cells are also highly active as they constantly receive and filter sensory input. Among the peripheral receptors, selective cell death has been well documented for hair cells, the mechanosensory cells of the mammalian inner ear that mediate hearing and balance. Auditory hair cell loss is a common feature of hearing impairment and can occur as a result of exposure to loud noise or clinical therapeutic compounds, and with aging (*Yang et al., 2015*). Hair cell loss occurs in a characteristic pattern relative

to both the frequency tuning of the cells and the cell type (inner vs. outer hair cells). After exposure to aminoglycoside antibiotics or with advancing age, for example, hair cells tuned to higher sound frequencies are lost before those tuned to lower frequencies. Similarly, outer hair cells are more susceptible to damage than inner hair cells (*Alharazneh et al., 2011*; *Mahendrasingam et al., 2011*; *Kamimura et al., 1999*; *Forge and Schacht, 2000*; *Forge and Richardson, 1993*; *Richardson and Russell, 1991*; *Kopke et al., 1999*). The cause of selective susceptibility for hair cells remains unclear, although some evidence suggests that differences in hair cell metabolism, free radical damage, or calcium ($Ca^{2+}$) handling may be contributors, thus implicating mitochondrial involvement (*Jensen-Smith et al., 2012*; *Sha et al., 2001*; *Engel et al., 2006*). A role for mitochondria is further supported by evidence of mitochondrial dysfunction during hair cell damage. Across species, dying hair cells exhibit swollen mitochondrial morphology and generate ROS in response to ototoxic agents, including aminoglycoside antibiotics and copper (*Owens et al., 2007*; *Fermin and Igarashi, 1983*; *Mangiardi et al., 2004*; *Lundquist and Wersäll, 1966*; *Bagger-Sjöbäck and Wersäll, 1978*; *Esterberg et al., 2013*; *Esterberg et al., 2014*; *Choung et al., 2009*; *Hirose et al., 1999*; *Olivari et al., 2008*; *Esterberg et al., 2016*).

In this study, we investigate the relationship between mitochondrial activity and selective toxicity of aminoglycosides using the zebrafish lateral line system. Sensory input from the lateral line is mediated by externally located clusters of mechanosensitive hair cells, called neuromasts. These cells allow fish to detect changes in water flow and to navigate their environment. They also share many similarities with hair cells of the inner ear (see *Nicolson, 2017* for review)(*Nicolson, 2017*). This includes conservation with human deafness genes as well as susceptibility to compounds that are ototoxic (*Whitfield, 2002*; *Nicolson, 2005*; *Harris et al., 2003*; *Ton and Parng, 2005*; *Ou et al., 2007*). Moreover, the surface location of the lateral line system provides a unique advantage in that we can monitor cellular changes that occur in vivo during physical or chemical manipulation with sub-cellular resolution. Like auditory hair cells, lateral line hair cells of older zebrafish are more susceptible to aminoglycoside-induced cell death (*Santos et al., 2006*). Unlike cochlear hair cells, lateral line hair cells do not exhibit intrinsic frequency selectivity and, although lateral line hair cells can be classified based on their polarity, they are not otherwise functionally sub-classified by type as are mammalian auditory and vestibular hair cells (*Kroese and Van Netten, 1989*; *Jiang et al., 2017*). For this reason, it has been particularly puzzling why some lateral line hair cells are more susceptible than others. Here we have imaged cumulative mitochondrial responses to hair cell stimulation or application of toxic aminoglycosides. We demonstrate that hair cell mechanotransduction (MET) activity has both acute and long-term effects on mitochondrial activity. Moreover, we demonstrate that cumulative changes in mitochondrial activity correspond with hair cell susceptibility to damage.

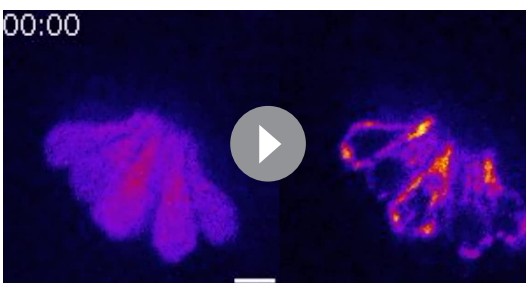

**Video 1.** Dynamic changes in hair cell cytoplasmic and mitochondrial $Ca^{2+}$ fluorescence with waterjet stimulation. Time-lapse video of a lateral line neuromast acquired from Tg[*myo6b:RGECO*]$^{vo10Tg}$ (left) and Tg[*myo6b:mitoGCaMP3*]$^{w119}$ (right) crossed fish. Hair cells were stimulated with a 10 Hz pressure wave, the timing of which is indicated (min:sec). Scale bar = 5 μm.

DOI: https://doi.org/10.7554/eLife.38062.021

## Results

### Mitochondria respond to acute hair cell stimulation

To examine mitochondrial responses to hair cell MET activity, we conducted in vivo time-lapse imaging studies of lateral line hair cells expressing genetically encoded $Ca^{2+}$ indicators. To visualize hair cell responses to stereocilia deflection, we used the Tg[*myo6b:RGECO*]$^{vo10Tg}$ line in which the red $Ca^{2+}$ indicator RGECO is cytoplasmically expressed in hair cells (referred to as cytoRGECO) (*Maeda et al., 2014*). We simultaneously monitored mitochondrial $Ca^{2+}$ in the same cells using a hair cell-specific, mitochondrially targeted GCaMP3 (Tg[*myo6b:mitoGCaMP3*]$^{w119}$) (referred to as mitoGCaMP) (*Esterberg et al., 2014*; *Esterberg et al., 2016*). Hair cells were imaged at a single plane and mechanically stimulated at

10 Hz using a pressure wave applied via a waterjet pipette. Both cytoRGECO and mitoGCaMP fluorescence intensity increased (*Video 1*), indicating that hair cell stimulation causes an influx of $Ca^{2+}$ into both the cytoplasm and mitochondria of the activated cells. Sample frames from four different videos are shown in *Figure 1* for cytoRGECO (panel A) and mitoGCaMP (panel B). These data demonstrate that mitochondria respond to hair cell stimulation. The two signals differ, however, in their kinetics. While cytoplasmic $Ca^{2+}$ levels increase and decrease rapidly with the onset and offset of waterjet stimulation, mitochondrial $Ca^{2+}$ levels exhibit a delayed rise and decay more slowly (*Figure 1C*). This is shown in the traces of $Ca^{2+}$ responses (*Figure 1C and D–F*) and in summary plots of rise times (*Figure 1D*; cyto vs mito: 8.9 ± 0.7 s vs 13 ± 0.9 s; mean ±SE; n = 21 cells). The integrated area (*Figure 1E*; cyto vs mito: 5.6 ± 0.7 vs 11 ± 1.3; mean ±SE; n = 19 cells) and peak change in the response (*Figure 1F*; cyto vs mito: 1.3 ± 0.04 vs 1.5 ± 0.06; mean ± SE; n = 19 vs 21 cells) were significantly greater for mitochondria than for cytoplasm.

## Mitochondrial activity in the absence of mechanotransduction

To assay acute mitochondrial activity another way, we used the cationic dye JC-1, a marker of mitochondrial membrane potential. JC-1 fluorescence shifts from green to red upon aggregation in energized mitochondria (*Smiley et al., 1991*; *Reers et al., 1991*). As a result, JC-1 provides a ratiometric readout of relative mitochondrial membrane potential as an indicator of mitochondrial activity. To assess whether MET activity altered mitochondrial activity, we examined JC-1 in wildtype or heterozygous larvae (WT/Het) and in *cadherin23* mutant larvae, also referred to as *sputnik*

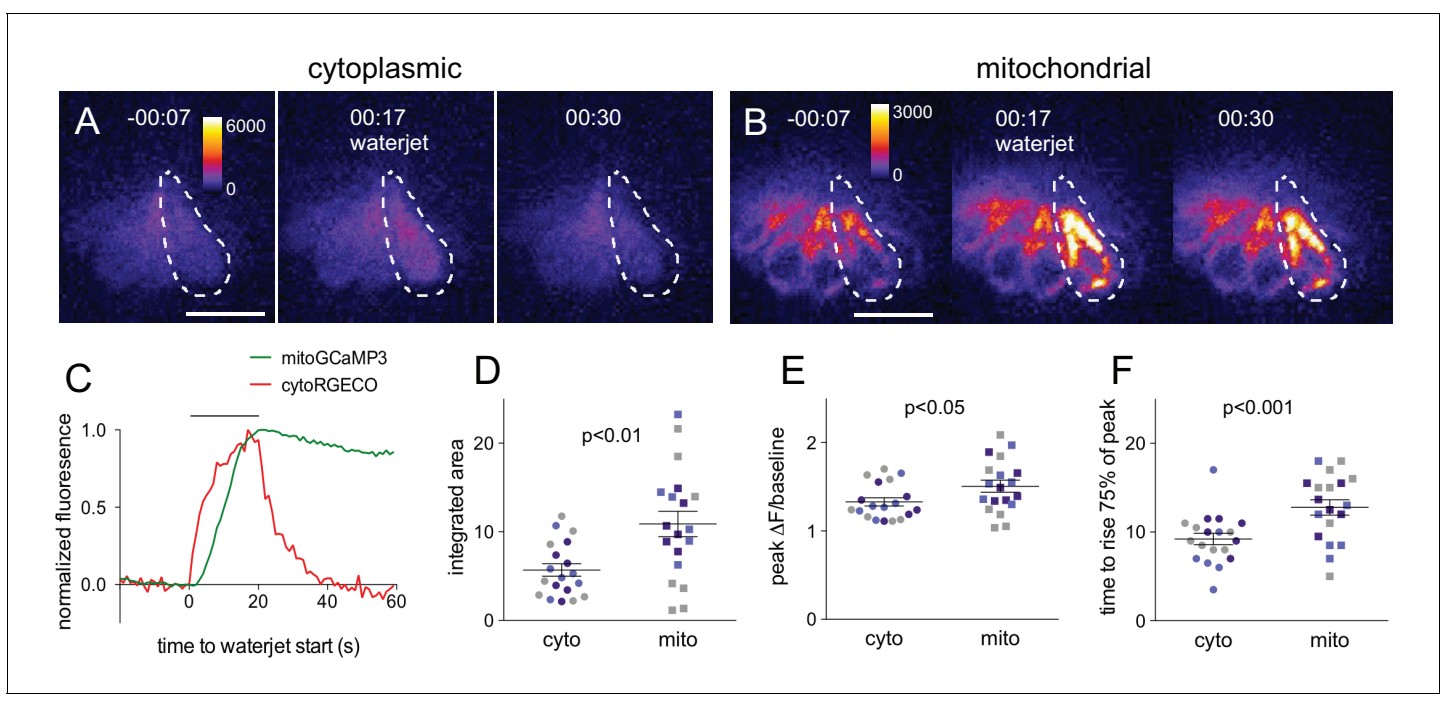

**Figure 1.** Mitochondrial $Ca^{2+}$ increases in response to hair cell stimulation. Frames from a time-lapse calcium imaging video acquired from Tg[*myo6b: RGECO*]$^{vo10Tg}$ fish (A) and Tg[*myo6b:mitoGCaMP3*]$^{w119}$ fish (B) during waterjet stimulation at 10 Hz. For each condition, fluorescence is shown before, during, and after stimulation (left, middle, right, respectively). (C) Normalized fluorescence of the example cell outlined by the dotted line in A and B. Hair cells were imaged from a lateral view, as indicated in the schematic shown in *Figure 1—figure supplement 1A*. Summary data of the integrated area (D), peak fluorescence (E), and rise time (F) for the cytoplasmic and mitochondrial calcium signal. Colored points indicate matched values from the same experiment. Values for (D): cyto vs. mito: 5.6 ± 0.7 vs 11 ± 1.3; mean ± SE; n = 19 cells; (E): cyto vs. mito: 1.3 ± 0.04 vs 1.5 ± 0.06; mean ± SE; n = 21 cells; (F): cyto vs. mito: 8.9 ± 0.7 s vs 13 ± 0.9 s; mean ± SE; n = 19 vs 21 cells. Cells were analyzed from 10 different fish across three different experiments. Two-tailed unpaired Student's t test was used to assess significance. Scale bar = 10 μm.

DOI: https://doi.org/10.7554/eLife.38062.002

The following figure supplement is available for figure 1:

**Figure supplement 1.** Orientation of imaging plane.

DOI: https://doi.org/10.7554/eLife.38062.003

mutants (*Nicolson et al., 1998*). As in mammalian hair cells, Cadherin23 is a critical component of the tip links necessary for function of the MET apparatus; thus *sputnik* mutant hair cells are MET inactive (*Söllner et al., 2004*). Example images of JC-1-labelled hair cells are shown in *Figure 2A and B*. JC-1 was analyzed as a ratio of red to green fluorescence intensity. Compared to age-matched WT/Het siblings, *sputnik* mutants exhibited significantly lower JC-1 fluorescence ratios, indicating that the mitochondria are depolarized (*Figure 2C*) (WT/Het: 0.25 ± 0.24 n = 8 fish, Mutant: 0.05 ± 0.07 n = 8 fish, Mann-Whitney U test, p < 0.01, mean ratio ± SD). These results suggest that mitochondrial activity is reduced in the absence of MET.

## Measuring mitochondrial aging and oxidation with mitoTimer

The waterjet and calcium imaging studies reveal that mitochondria respond to hair cell MET activity. $Ca^{2+}$ flux can have multiple effects on mitochondrial function, including regulation of electron transport during oxidative phosphorylation (OXPHOS) and generation of ROS (*Brookes et al., 2004*). We next wanted to examine whether acute MET activity causes persistent effects on the state of mitochondria. To look at cumulative mitochondrial activity over time, we used a transgenic zebrafish expressing the reporter mitoTimer in all hair cells (Tg[*myo6b:mitoTimer*]$^{w208}$; here referred to as mitoTimer). mitoTimer encodes a DsRed mutant (DsRed1-E5) with a COXVIII mitochondrial targeting sequence (*Terskikh et al., 2000*). While newly synthesized mitoTimer fluoresces green, the fluorescence shifts irreversibly to red over time due to dehydrogenization of the Tyr-67 residue (*Verkhusha et al., 2004*; *Yarbrough et al., 2001*). The change in fluorescence spectra and the means by which it occurs has made mitoTimer a useful tool to examine mitochondrial turnover and transport, as well as reporting cumulative redox and oxygenation history (*Ferree et al., 2013*; *Hernandez et al., 2013*; *Laker et al., 2014*; *Stotland and Gottlieb, 2016*; *Laker et al., 2017*; *Wilson et al., 2019*; *Lidsky et al., 2018*). We used mitoTimer to first examine changes in the state of mitochondria during neuromast maturation. mitoTimer signal was measured at 3 days post-fertilization (dpf), an early time point in lateral line hair cell maturation, through 5dpf, the point at which the neuromast is considered functionally mature. Example images of mitoTimer-expressing hair cells are shown in *Figure 3A and B*. mitoTimer fluorescence, reported as a ratio of red to green, increases significantly between 3dpf and 4dpf, as well as between 4dpf and 5dpf (*Figure 3C*) (3dpf: 1.03 ± 0.05, 4dpf: 2.47 ± 0.66, 5dpf: 7.58 ± 1.2; Kruskal-Wallis test, Dunn's post-test, p < 0.001; mean ratio ±SD; n = 15–18 fish). The ratio shift corresponds with the timing of functional maturation of the neuromast, including increased hair cell MET activity. These observations suggest that mitochondrial activity increases with cumulative hair cell activity. mitoTimer is a multifaceted indicator that can reflect multiple aspects of mitochondrial activity and metabolism, including oxidation, over

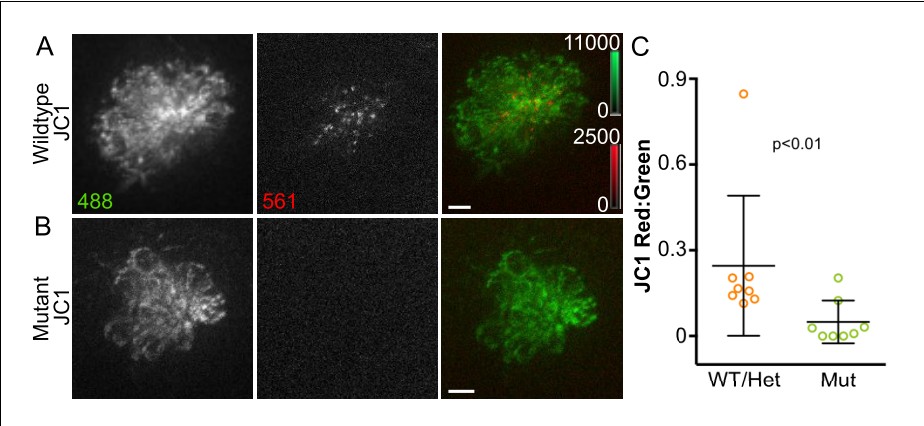

**Figure 2.** Acute mitochondrial activity is reduced in the absence of MET. (A, B) Maximum projections of hair cells from WT/Het and *sputnik* mutant siblings incubated in JC-1 dye. Hair cells were imaged from a dorsal view, as indicated in the schematic shown in *Figure 1—figure supplement 1B*. (C) Mean JC-1 fluorescence plotted as a ratio of red:green. WT/Het: 0.25 ± 0.24 n = 8 fish; Mutant: 0.05 ± 0.07 n = 8 fish; mean ratio ± SD. Mann-Whitney U test was used to assess significance. Value for each fish represents the mean of 3 neuromasts. Scale bar = 5 μm.
DOI: https://doi.org/10.7554/eLife.38062.004

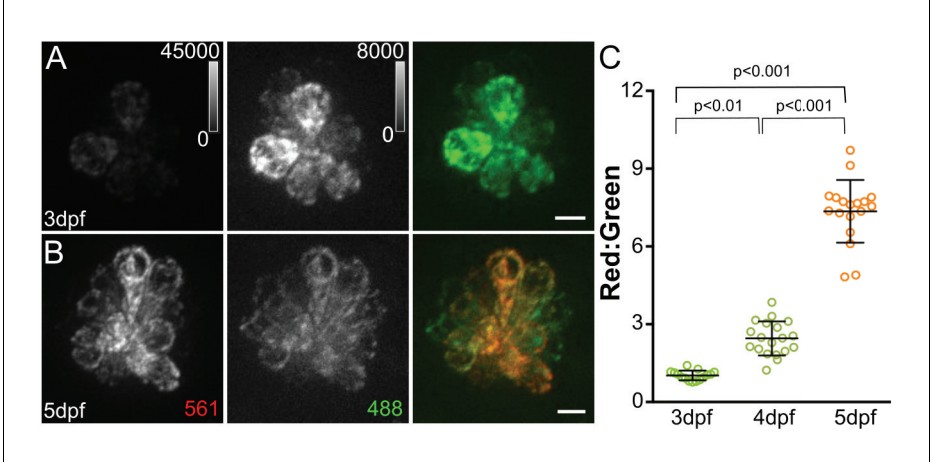

**Figure 3.** MitoTimer fluorescence ratio increases with neuromast maturation. Maximum projections of hair cells from Tg[*myo6b:mitoTimer*]$^{w208}$ fish at 3dpf (**A**) and 5dpf (**B**). (**C**) Mean mitoTimer fluorescence plotted as ratio of red:green at 3, 4, and 5dpf. 3dpf: 1.03 ± 0.05, 4dpf: 2.47 ± 0.66, 5dpf: 7.58 ± 1.2; mean ratio ±SD; n = 15–18 fish. Significance was analyzed by Kruskal-Wallis test with Dunn's post-test. Value for each fish represents the mean of 2–3 neuromasts. Scale bar = 5 μm.
DOI: https://doi.org/10.7554/eLife.38062.005

time. During hair cell maturation, two factors could contribute to changes in mitoTimer fluorescence: the age of the cell and rate of oxidation in the organelle. Because some hair cells are added to the neuromast during development from 3-5dpf, differences in the mitoTimer ratio may reflect the relative age of hair cells in the cluster. To test this idea, we incubated free-swimming larvae in the nuclear dye Hoechst, which differentially labels mechanotransducing hair cells (see *Figure 4—figure supplement 1*), for 30 min at 3dpf. This is an intermediate developmental time point relative to neuromast functional maturity, where only some hair cells have active MET and additional hair cells are still being added to the cluster. At 4dpf, we measured the ratio of mitoTimer fluorescence (red: green) in individual cells, comparing older (Hoechst labeled) to younger (unlabeled) hair cells. *Figure 4A* shows an example image of Hoechst staining in mitoTimer-expressing hair cells 24 hr after labeling. To directly compare hair cells within the same neuromast, we normalized the mito-Timer ratio for each individual cell to the median ratio of the neuromast as a whole. We found that younger hair cells lacking Hoechst labeling have a mitoTimer ratio near or below the median (*Figure 4B*). The opposite is true for older hair cells that are positive for Hoechst labeling after 24 hr (Hoechst positive: 1.71 ± 0.52, n = 45 cells; Hoechst negative: 0.80 ± 0.45, n = 80 cells; Mann-Whitney U test, p < 0.001 ; mean ratio (normalized) ±SD; five fish per group). We confirmed that Hoechst labeling itself does not increase the mitoTimer fluorescence ratio by comparing the distribution of single cell red:green fluorescence from zebrafish treated with Hoechst to an untreated group (*Figure 4—figure supplement 2*). Together, these experiments suggest that cells with increased mito-Timer ratios are older and, as a result, have also been capable of MET activity for a longer period of time.

## Blocking mechanotransduction has long-term effects on mitochondria

We next tested whether long-term changes in MET activity would result in alterations in mitochondrial activity, reflected by changes in mitoTimer fluorescence. We first assayed this in the absence of hair cell stimulation by measuring mitoTimer fluorescence in *sputnik* mutants. Compared to age-matched WT/Het siblings, the mitoTimer fluorescence ratio was significantly decreased in *sputnik* mutants, with a difference of 66.3% (*Figure 5A–C*; Mann-Whitney U test, p < 0.001; n = 14 WT/Het fish and 15 mutant fish; data combined from three experiments). Similar results were obtained when embryos were incubated in the MET-blocking drug benzamil (200 μM) for 48 hr (3-5dpf; *Figure 5D–F*) (*Hailey et al., 2017*). We observed a reduction of 43.4% relative to 0.5% DMSO control (Mann-Whitney U test, p < 0.001; n = 17 fish per group; data combined from two experiments).

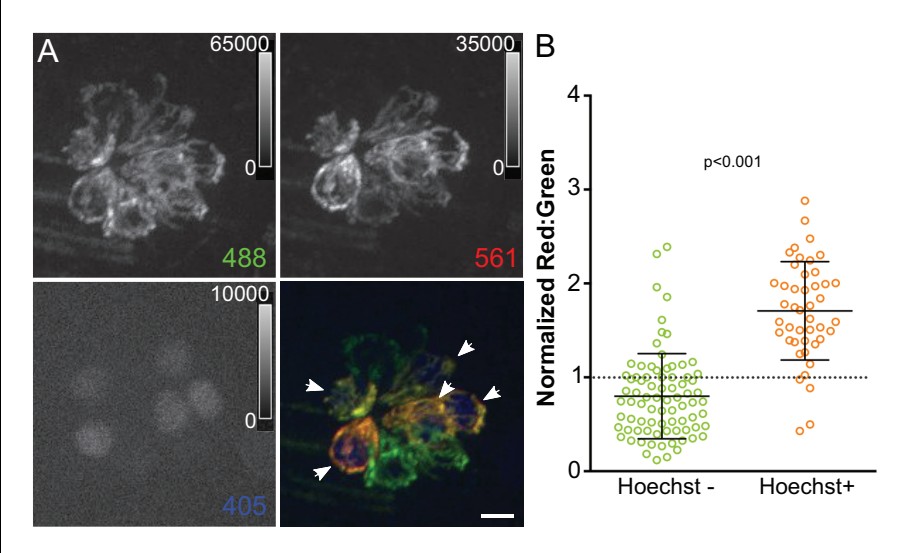

**Figure 4.** Mitochondrial oxidation corresponds with hair cell age and mechanotransduction activity. (**A**) Maximum projection of mitoTimer expressing hair cells co-labeled with Hoechst. Hair cells were imaged at 4dpf, 24 hr after Hoechst treatment. Arrowheads mark Hoechst-positive cells. (**B**) mitoTimer fluorescence ratio for cells that are positive for Hoechst staining compared to Hoechst negative cells. mitoTimer ratios are normalized to the median, which is indicated by the dotted line. Hoechst positive: 1.71 ± 0.52, n = 45 cells; Hoechst negative: 0.80 ± 0.45, n = 80 cells; mean ratio (normalized) ±SD; five fish, three neuromasts per fish. Mann-Whitney U test was used to assess significance. Scale bar = 5 μm.

DOI: https://doi.org/10.7554/eLife.38062.006

The following figure supplements are available for figure 4:

**Figure supplement 1.** Hoechst nuclear labeling is MET-dependent.
DOI: https://doi.org/10.7554/eLife.38062.007
**Figure supplement 2.** Hoechst incubation does not affect mitoTimer ratio compared to control.
DOI: https://doi.org/10.7554/eLife.38062.008

---

Since mitoTimer fluorescence shifts with cell age (*Figure 4*), the decreased red:green fluorescence observed in the absence of MET could potentially indicate that hair cells within these neuromasts were younger in age. This could arise due to differences hair cell turnover. To investigate the effect of reduced MET on hair cell turnover, we conducted photoactivation experiments with a transgenic line expressing nuclear-localized, photoconvertible Eos under control of the *atoh1a* locus (Tg [*Atoh1a:nls-Eos*]^w214^), with or without benzamil treatment. Under normal conditions, Eos protein fluoresces green, but undergoes an irreversible conformational change following UV light exposure, which converts its emitted fluorescence to red. Both support cells and hair cells are labelled in the Tg[*Atoh1a:nls-Eos*]^w214^ line. As a result, fish were crossed to Tg[*Pou4f3:gap43-GFP*] to distinguish hair cell and support cell nuclei based on hair cell-specific membrane GFP expression. Nuclear-localized Eos was photoconverted prior to incubation in benzamil (200 μM) for 48 hr. Control and benzamil-treated fish were imaged at 5dpf to identify label-retaining nuclei relative to the total number of hair cells in the neuromast. The ratio of label retaining hair cells was consistent in both control and benzamil treated conditions, indicating that hair cell turnover is unchanged in the absence of MET activity (*Figure 5—figure supplement 1*; Control: 0.72 ± 0.12, n = 8 control fish; Treated: 0.70 ± 0.14, n = 10 fish).

While changes in mitoTimer fluorescence reflect mitochondrial oxidation, they might also reflect rates of mitochondrial protein import or turnover. To investigate the effect of hair cell activity on mitochondrial turnover, we conducted photoactivation experiments with mitochondrially-localized Eos, Tg[*myo6b:mitoEos*]^w207^, in WT/Het and *sputnik* mutants. Mitochondrially-localized Eos in hair cells was photoconverted at 5dpf and then monitored to measure loss of red fluorescence as an estimate of mitochondrial or mitochondrial protein turnover. Mitochondrial fluorescence was measured

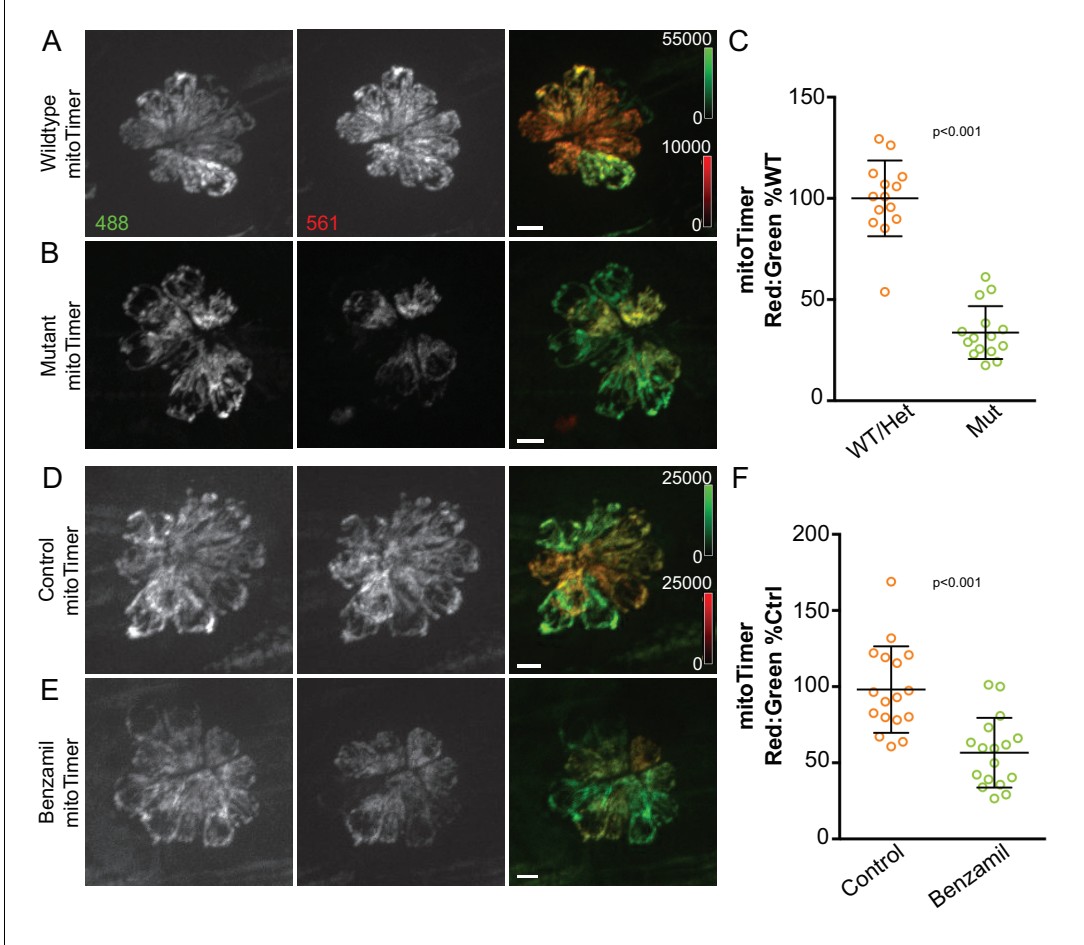

**Figure 5.** Mitochondrial activity depends on hair cell mechanotransduction. (**A, B**) Maximum projections of hair cells from WT/Het and *sputnik* mutant siblings crossed to Tg[*myo6b:mitoTimer*]^w208. (**C**) mitoTimer mean fluorescence ratio for WT/Het and *sputnik* larvae. WT/Het: 100 ± 18.7, n = 14 fish; Mutant: 34 ± 13.1,15 mutant fish; mean (% WT/Het)±SD. Value for each fish represents the mean of 2–4 neuromasts. (**D, E**) Maximum projections of mitoTimer-expressing hair cells from control larvae and larvae incubated in 200 µM benzamil. (**F**) mitoTimer mean fluorescence ratio for larvae incubated in benzamil compared to DMSO control. Control: 100 ± 28.7; Treated: 56.6 ± 22.7, n = 17 fish per group; mean (% Control)±SD. Value for each fish represents the mean of 2–4 neuromasts. Mann-Whitney U test was used to assess significance. All larvae imaged at 5dpf. Scale bar = 5 µm.
DOI: https://doi.org/10.7554/eLife.38062.009

The following figure supplement is available for figure 5:

**Figure supplement 1.** Reduced hair cell activity does not influence hair cell turnover.
DOI: https://doi.org/10.7554/eLife.38062.010

at 8 and 16 hr post-photoconversion. *Figure 6* shows the change in fluorescence at these time points. While green fluorescence remained consistent (data not shown), the percent decrease in red fluorescence was 38.9% and 75.5% for wildtype and mutants, respectively (Mann-Whitney U test, p < 0.001; n = 9 fish per group). The change in red fluorescence over time indicates that mitochondrial or protein turnover increases with reduced hair cell activity.

## Sustained mechanical stimulation results in cumulative mitochondrial changes

We next asked if sustained hair cell stimulation increases mitoTimer fluorescence. To increase hair cell stimulation, water currents were generated around the larvae by placing them on an orbital shaker at 60 RPM for 24 hr at room temperature. While orbital rotation significantly increased the mitoTimer fluorescence ratio in WT/Het fish, a statistically significant difference was not observed in age-matched *sputnik* mutants (WT/Het Control: 100 ± 34 n = 47 fish, WT/Het Orbital shaker:

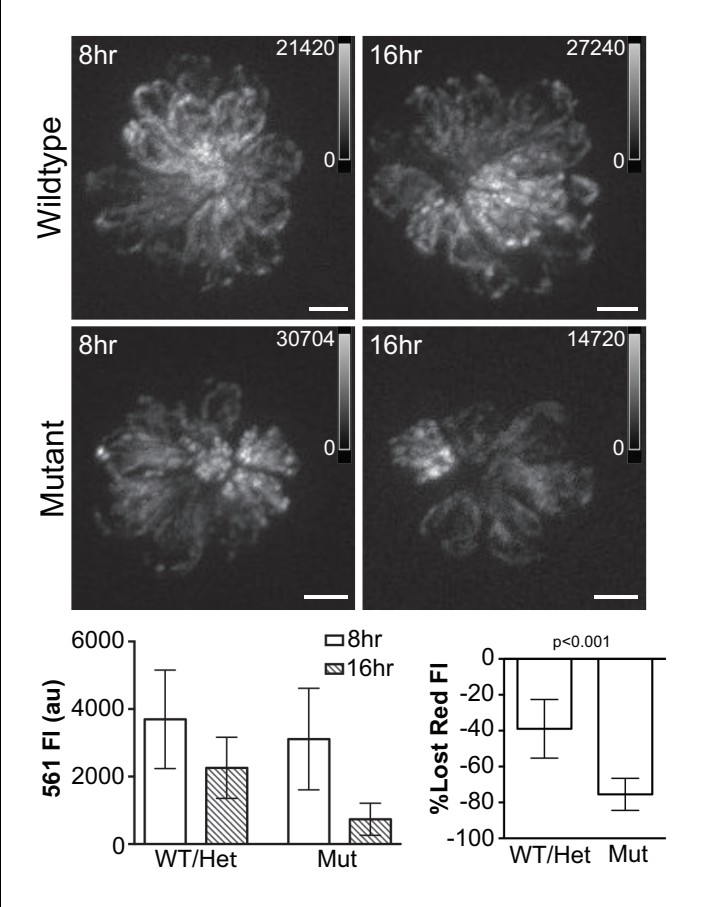

**Figure 6.** Hair cell activity influences mitochondrial turnover. (**A, B**) Maximum projections of mitoEos red fluorescence 8 and 16 hr after photoconversion in wildtype and *sputnik* fish crossed to Tg[*myo6b:mitoEos*]^w207. (**C**) Mean red fluorescence decreases over time in both wildtype and mutant fish. WT/Het: 8 hr, 3696 ± 1083; 16 hr, 2257 ± 604; Mutant: 8 hr, 2942 ± 878; 16 hr, 721 ± 262, mean ± SD. (**D**) The loss of red fluorescence is significantly greater in mutant animals. WT/Het: 38.9 ± 16.4; Mutant: 75.5 ± 8.9; mean (% lost)±SD. n = 9 fish per group. Value for each fish represents the mean of 3 neuromasts. Significance analyzed by Mann-Whitney U test. Scale bar = 5 μm.

DOI: https://doi.org/10.7554/eLife.38062.011

128 ± 47 n = 48 fish, Mutant Control: 58 ± 24 n = 27 fish, Mutant Orbital shaker: 70 ± 21 n = 28 fish; Kruskal-Wallis test, Dunn's post-test, p < 0.05; mean ratio reported as a percent of WT/Het Control ± SD; data combined from four experiments) (*Figure 7A*, *Figure 7—figure supplement 1*). Incubation on the orbital shaker for an additional 24 hr resulted in further increase in the red:green ratio (*Figure 7—figure supplement 2*).

The results thus far demonstrate that mitoTimer fluorescence appears to shift with both hair cell age (*Figure 4*) and activity (*Figures 5* and *7A*). To tease this apart, we next paired Hoechst nuclear labeling with orbital rotation to measure stimulation-induced fluorescence shifts in similarly aged cells. 4dpf larvae were incubated in Hoechst for 30 min prior to orbital stimulation. After 24 hr, we measured the mitoTimer fluorescence in individual cells, comparing older (Hoechst labeled) and younger (unlabeled) hair cells in control and stimulated conditions. We observed that orbital stimulation significantly increased the red:green fluorescence of older, Hoechst-positive hair cells compared to control (*Figure 7B*). Alternatively, there was no significant difference in the mitoTimer fluorescence ratios of younger, Hoechst-negative cells in both conditions (Hoechst-positive Control: 1.36 ± 0.37, n = 94 cells; Hoechst-negative Control: 0.90 ± 0.22, n = 52 cells; Hoechst-positive Orbital Shaker: 1.70 ± 0.47, n = 105 cells; Hoechst-negative Orbital Shaker: 0.97 ± 0.23, n = 77 cells; mean ratio ± SD; Kruskal-Wallis test, Dunn's post-test, p < 0.001; 10 fish per group). These results

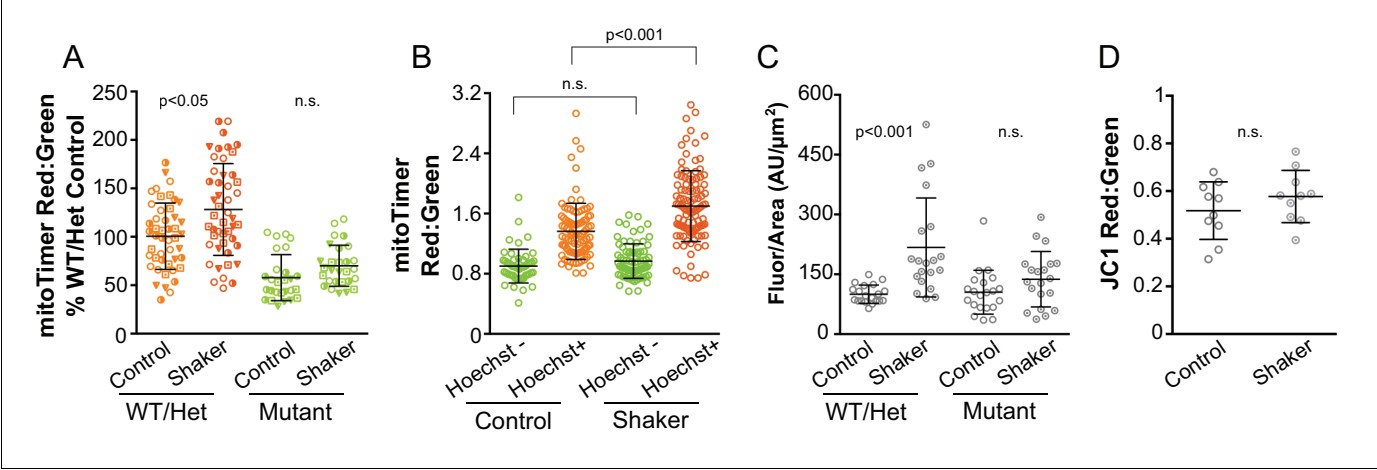

**Figure 7.** Sustained hair cell stimulation through orbital shaking increases cumulative mitochondrial activity and hair cell oxidation. (**A**) mitoTimer fluorescence ratio measured from hair cells in WT/Het and *sputnik* mutants following 24 hr of orbital shaking. Symbols indicate matched values from the same experiment. WT/Het Control: 100 ± 34 n = 47 fish; WT/Het Orbital shaker: 128 ± 47 n = 48 fish; Mutant Control: 58 ± 24 n = 27 fish; Mutant Orbital shaker: 70 ± 21 n = 28 fish; mean (%WT/Het Control) ± SD. (**B**) mitoTimer fluorescence ratio for cells that are positive for Hoechst staining compared to Hoechst negative cells following 24 hr of orbital stimulation. Fluorescence was measured at 5dpf, 24 hr after Hoechst treatment and orbital rotation. Hoechst-positive Control: 1.36 ± 0.37, n = 94 cells; Hoechst-negative Control: 0.90 ± 0.22, n = 52 cells; Hoechst-positive Orbital Shaker: 1.70 ± 0.47, n = 105 cells; Hoechst-negative Orbital Shaker: 0.97 ± 0.23, n = 77 cells; mean ratio ± SD; 10 fish per group. (**C**) CellROX green fluorescence measured from hair cells in WT/Het and *sputnik* mutants following 24 hr of orbital shaking. WT/Het Control: 100 ± 23 n = 20 fish, WT/Het Orbital shaker: 217 ± 124 n = 20 fish, Mutant Control: 105 ± 55 n = 21 fish, Mutant Orbital shaker: 138 ± 58 n = 20 fish; mean (%WT/Het Control) ± SD. (**D**) JC-1 fluorescence ratio measured from hair cells following 24 hr of orbital stimulation. Control: 0.52 ± 0.12; Orbital Shaker: 0.58 ± 0.11; n = 10 fish per group; Mann-Whitney U test, p = 0.32. Values for each fish represent the mean of 2–3 neuromasts. Significance analyzed by Kruskal-Wallis test with Dunn's post-test. All larvae imaged at 5dpf.

DOI: https://doi.org/10.7554/eLife.38062.012

The following figure supplements are available for figure 7:

**Figure supplement 1.** Mito Timer fluorescence ratio measured from hair cells in WT/Het and *sputnik* mutants following 24 hr of orbital shaking.
DOI: https://doi.org/10.7554/eLife.38062.013

**Figure supplement 2.** Changes in mitoTimer fluorescence ratios after orbital shaking is time-dependent.
DOI: https://doi.org/10.7554/eLife.38062.014

**Figure supplement 3.** Hair cell oxidation increases with sustained stimulation via orbital shaking.
DOI: https://doi.org/10.7554/eLife.38062.015

**Figure supplement 4.** Mitochondrial polarization does not shift with short-term hair cell stimulation.
DOI: https://doi.org/10.7554/eLife.38062.016

suggest that the hair cells most affected by shaker stimulation are those that are already active before the onset of stimulation.

Changes in mitoTimer fluorescence caused by increased MET activity may reflect changes in mitochondrial oxidation (*Laker et al., 2014*; *Wilson et al., 2019*). We therefore examined hair cells labeled with the ROS-indicator dye CellROX green in combination with orbital rotation. CellROX is a cumulative fluorescent dye that localizes to nuclei, exhibiting weak fluorescence that increases in brightness upon oxidation. With longer incubation times, CellROX labels both hair cell and support cell nuclei (*Figure 7—figure supplement 3*). Hair cell fluorescence was measured in a single image plane selected to maximize the hair cell nuclei visualized and avoid signal from support cells. Hair cells were distinguished based on their nuclear shape and apical position within the neuromast (*Figure 7—figure supplement 3*). Similar to the mitoTimer fluorescent shift, orbital shaking significantly increased CellROX fluorescence in WT/Het fish (*Figure 7C*) (WT/Het Control: 100 ± 23 n = 20 fish, WT/Het Orbital shaker: 217 ± 124 n = 20 fish, Mutant Control: 105 ± 55 n = 21 fish, Mutant Orbital shaker: 138 ± 58 n = 20 fish; Kruskal-Wallis test, Dunn's post-test, p < 0.001; mean ratio reported as a percent of WT/Het Control ± SD; data combined from three experiments). These results

demonstrate that cumulative mitochondrial activity and hair cell oxidation are influenced by long-term changes in MET activity.

To examine whether changes in instantaneous mitochondrial activity could be detected with sustained hair cell stimulation, 4dpf fish were stimulated via orbital shaking for 24 hr and then incubated in JC-1. Unlike the cumulative mitochondrial and oxidative indicators, analysis of JC-1 labelling did not reveal persistent changes in mitochondrial polarity (Control: 0.52 ± 0.12, n = 10 fish; Orbital Shaker: 0.58 ± 0.11, n = 10 fish; mean ratio ±SD; Mann-Whitney U test, p = 0.32) (*Figure 7D*). Similarly, no difference in the fluorescence ratio was observed when hair cells were pre-labeled with JC-1 before a shorter period of orbital shaking (90 min) (*Figure 7—figure supplement 4*).

## History of mitochondrial activity predicts likelihood of hair cell susceptibility to damage

Like mammalian auditory and vestibular hair cells, zebrafish lateral line hair cells are highly susceptible to damage caused by aminoglycoside antibiotics and become oxidized following drug exposure. Mitochondria play a key role, as a recent zebrafish study identified mitochondria as the source of ROS during neomycin-induced hair cell death following mitochondrial $Ca^{2+}$ elevation (*Esterberg et al., 2016*). Moreover, mitigating mitochondrial ROS generation and $Ca^{2+}$ uptake confers enhanced hair cell protection. Previous studies have also noted selective and asynchronous hair cell death as a result aminoglycoside exposure, as well as variation in levels of ROS and $Ca^{2+}$ in individual cells. The underlying causes of this differential vulnerability remain puzzling, particularly because aminoglycoside exposure is comparable for all hair cells in a neuromast (*Hailey et al., 2017*).

Since mitochondrial $Ca^{2+}$ influx and oxidation also occur in response to hair cell stimulation, we sought to determine whether acute hair cell and mitochondrial activity correspond with selective susceptibility to damage following neomycin treatment. To do so, time-lapse imaging was conducted with fish labeled with the mitochondrial polarity indicator dye, JC-1. Since the JC-1 fluorescence ratio decreases in the absence of hair cell MET activity, measuring the baseline fluorescence of JC-1 provides an estimate of mitochondrial activity just prior to neomycin exposure. Baseline fluorescence was recorded prior to treatment and animals were then exposed to 50 μM neomycin, a concentration that leads to approximately 40% hair cell death (*Harris et al., 2003*). Cells were subsequently categorized as alive or dead based on observed fragmentation and clearance from the neuromast (*Esterberg et al., 2013*). We observed no difference in the mean fluorescence ratios of living and dying cells using this measure (Living cells: 0.13 ± 0.14, n = 46; dying cells: 0.12 ± 0.11, n = 35; mean ratio ±SD; six fish, 2–3 neuromasts per fish; Mann-Whitney U test, p = 0.60) (*Figure 8*). These results demonstrate that acute differences in mitochondrial activity just prior to neomycin exposure do not correspond with likelihood of subsequent hair cell death. We also examined whether variation in baseline cytoplasmic or mitochondrial calcium indicator fluorescence corresponded with neomycin susceptibility. Hair cell-specific mitoG-CaMP3 and cytoRGECO baseline fluorescence were recorded prior to treatment, with values across cells reflecting an 8-fold range and an 11-

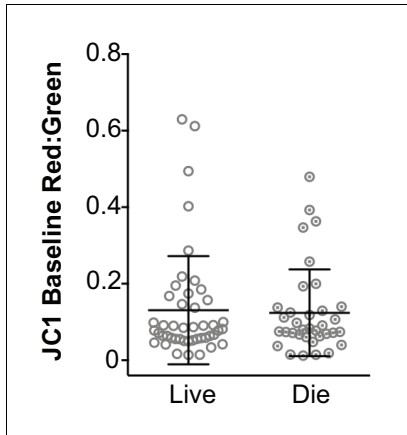

**Figure 8.** Acute mitochondrial activity does not correspond with likelihood of hair cell death in response to 50 μM neomycin exposure. Baseline mean fluorescence ratio of JC-1 for living and dying cells. Living cells: 0.13 ± 0.14, n = 46; dying cells: 0.12 ± 0.11, n = 35; mean ratio ± SD; six fish, 2–3 neuromasts per fish; Mann-Whitney U test, p = 0.60. Fluorescence measurements were taken just prior to neomycin treatment.

DOI: https://doi.org/10.7554/eLife.38062.017

The following figure supplement is available for figure 8:

**Figure supplement 1.** Baseline calcium levels do not correspond with likelihood of hair cell death in response to 50 μM neomycin exposure.

DOI: https://doi.org/10.7554/eLife.38062.018

fold range, respectively. Consistent with previous findings, we found no difference in the fluorescence values of living and dying cells for both mitoGCaMP3 and cytoRGECO (*Figure 8—figure supplement 1*) (*Esterberg et al., 2013*).

We next examined whether differences in mitoTimer fluorescence, reflecting cumulative changes in mitochondrial activity, would correlate with susceptibility to damage. We measured mitoTimer fluorescence by live imaging of hair cells before and during neomycin treatment (*Figure 9A*; *Video 2*). Baseline ratios of mitoTimer for individual cells were normalized to the median ratio of all cells in the corresponding neuromast. We observed that the average mitoTimer ratio for surviving hair cells was below the median ratio at baseline, while the average ratio for cells that die following treatment was above the median (*Figure 9B*) (Live: 0.77 ± 0.37, Die: 1.30 ± 0.34; Mann-Whitney U test, $p < 0.001$; mean ratio (normalized) ±SD; n = 142 living cells, 74 dying cells from six fish; data combined from two experiments). Comparing the two distributions, a surviving cell had approximately an 85% chance of having a lower red:green ratio than a dying cell. We also calculated the degree to which classifying cells as red (red:green above median) or green (red:green below median) was effective in prospectively identifying those that would live or die. Using this classification, 58 of 74 dying cells were redder, while only 42 of 142 living cells were so ($p < 0.0001$, Fisher Exact Test). This distribution shows that the red-green classification scheme provides a sensitivity of 0.78, specificity of 0.70, and a diagnostic odds ratio of 8.63, reflecting strong predictive value. In this case, the diagnostic odds ratio indicates that red cells are over eight-fold more likely to die than live after neomycin exposure.

Because mitoTimer ratios reflect both age and activity, we attempted to dissociate these variables by labeling older hair cells with Hoechst dye and comparing survival of older and younger cells along with mitoTimer signal after neomycin treatment. We labeled fish expressing mitoTimer with Hoechst dye at four dpf, and then treated them with 50 μM or 100 μM neomycin 24 hr later. For both older (Hoechst+) and younger (Hoechst-) cells, we saw a dose dependent decrease in hair cell survival (*Figure 10A*; Control Hoechst+: 8.88 ± 1.83; 50 μM Neo Hoechst+: 2.08 ± 1.68; 100 μM Neo Hoechst+: 0.35 ± 0.65; Control Hoechst-: 3.63 ± 1.58; 50 μM Neo Hoechst-: 3.16 ± 1.60; 100 μM Neo Hoechst-: 2.00 ± 1.00; mean cells/neuromast ± SD; nine fish per group). Older cells were more sensitive to 50 μM neomycin compared to younger cells, as we reported previously (*Santos et al., 2006*). We next compared mitoTimer fluorescence of surviving cells in the treated groups to those in control animals that were not exposed to neomycin (*Figure 10B*). For surviving hair cells – including both older hair cells, labeled with Hoechst, and younger unlabeled cells – the mean red:green ratio was significantly lower than untreated controls (Control Hoechst+: 1.03 ± 0.25 n = 124 cells; 50 μM Neo Hoechst+: 0.78 ± 0.23, n = 38 cells; 100 μM Neo Hoechst+: 0.76 ± 0.25, n = 8 cells; Control Hoechst-: 0.50 ± 0.09, n = 52 cells; 50 μM Neo Hoechst-: 0.45 ± 0.14, n = 64 cells; 100 μM Neo Hoechst-: 0.42 ± 0.12, n = 46 cells; mean ratio ± SD; nine fish per group). These results suggest that cells with a higher mitoTimer ratio in both older and younger populations are more susceptible to damage. Together with our previous results, this suggests that the most susceptible cells are the older and historically MET active cells in the cluster. In this way, the cumulative history of mitochondrial activity in hair cells predicts their vulnerability to neomycin exposure.

## Discussion

By monitoring lateral line hair cells using live-imaging techniques, we were able to uncover effects of hair cell MET activity on mitochondria in an intact organism. With acute stimulation, hair cells exhibit increased cytosolic and mitochondrial $Ca^{2+}$ influx with distinct kinetic properties. With chronic changes to hair cell MET activity, we observed corresponding changes in mitoTimer fluorescence, indicative of altered mitochondrial oxidation and turnover. In support of this interpretation, we also observed activity-dependent changes in mitochondrial polarization as well as hair cell oxidation. Our ability to visualize the chronic impact of hair cell activity on mitochondria allowed us to identify hair cells that are more susceptible to aminoglycoside-induced damage. Our findings show that the cumulative history of mitochondrial activity, which may be influenced by age and MET activity, is predictive of differential hair cell death.

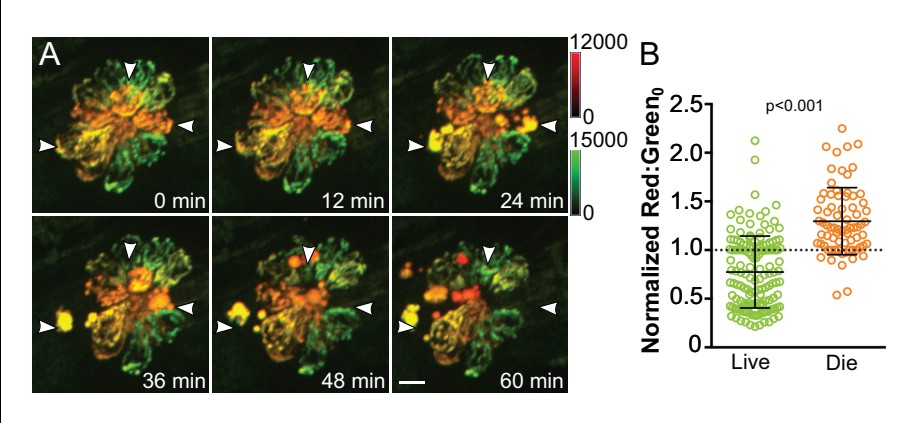

**Figure 9.** Cumulative mitochondrial activity reflects the likelihood of hair cell death following neomycin-induced damage. (**A**) Frames from a time-lapse imaging video acquired from Tg[*myo6b:mitoTimer*][w208] fish treated with 50 µM neomycin. Images are maximum projections. Arrowheads indicate dying cells. (**B**) Baseline mitoTimer fluorescence ratio for living and dying cells following neomycin exposure. mitoTimer ratios are normalized to the median, which is indicated by the dotted line. Live: 0.77 ± 0.37, n = 142 cells; Die: 1.30 ± 0.34, n = 74 cells; mean ratio (normalized) ±SD; six fish, 2–3 neuromasts per fish. Mann-Whitney U test was used to assess significance. Scale bar = 5 µm.

DOI: https://doi.org/10.7554/eLife.38062.019

## Mitochondrial metabolism and dynamics during normal cellular activity

Mitochondria are critical organelles, generating ATP through OXPHOS, while also playing an important role in $Ca^{2+}$ homeostasis and production of ROS. These mitochondrial activities are all interrelated. ROS production within mitochondria occurs as a result of normal metabolic activity at two sites within the electron transport chain, complexes I and III. Electron leakage at these complexes leads to the generation of superoxide, which is converted into membrane-permeable hydrogen peroxide (*Loschen et al., 1974*; *Turrens, 2003*). Mitochondrial $Ca^{2+}$ signaling influences this process as it can stimulate OXPHOS, and therefore, electron flow (*Brookes et al., 2004*).

The relationship between $Ca^{2+}$, metabolic activity, and ROS production is particularly notable for hair cells, where constant stimulation and sensory input leads to $Ca^{2+}$ influx. Increased mitochondrial $Ca^{2+}$ levels were also observed with MET activity, demonstrating that hair cell activity directly affects mitochondria. Compared to cytoplasmic $Ca^{2+}$, the larger integrated area and rise time of the mitochondrial signal (*Figure 1D,F*) are consistent with the idea that mitochondria may act as $Ca^{2+}$ buffers, mediating $Ca^{2+}$ homeostasis in addition to other means of cellular $Ca^{2+}$ extrusion. This role for hair cell mitochondria has been postulated in mammalian hair cells based, in part, on apical localization of the organelles and through $Ca^{2+}$-imaging studies during transduction (*Weaver and Schweitzer, 1994*; *Rüsch et al., 1998*; *Beurg et al., 2010*). Mitochondrial $Ca^{2+}$ buffering could be further investigated through manipulation of $Ca^{2+}$ uptake and release mechanisms during hair cell stimulation.

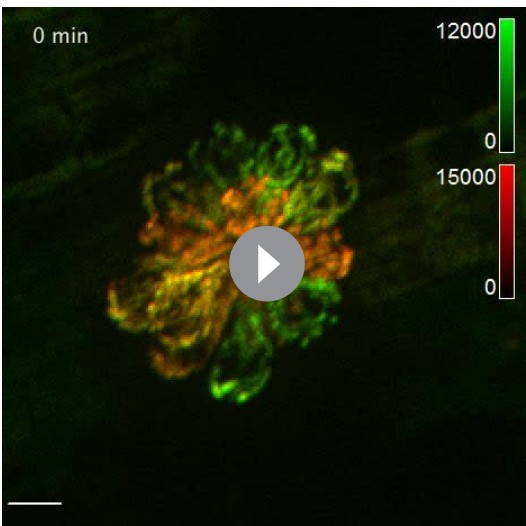

**Video 2.** Differential cell death after low-dose neomycin exposure among mitoTimer-expressing hair cells. Time-lapse video of a lateral line neuromast from a Tg[*myo6b:mitoTimer*][w208] fish exposed to 50 µM neomycin. Still frames from this time-laspe video are shown in *Figure 9*. Scale bar = 5 µm.

DOI: https://doi.org/10.7554/eLife.38062.022

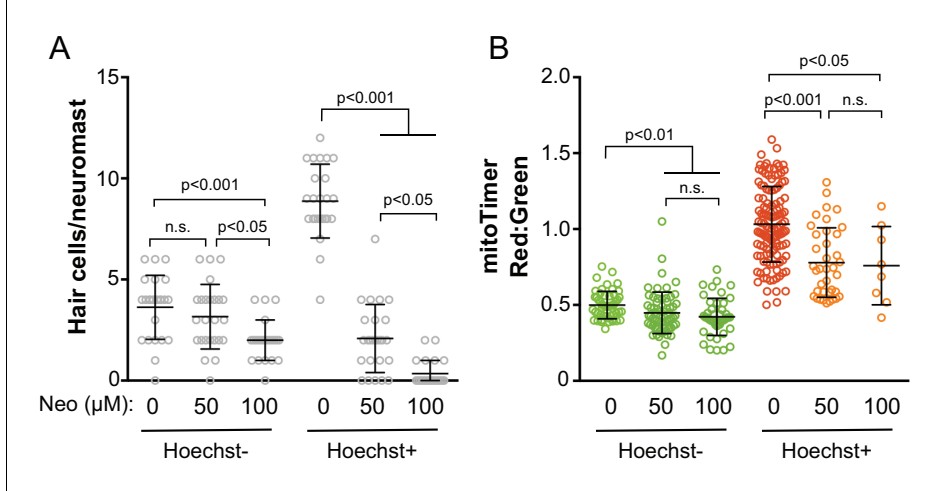

**Figure 10.** Older and historically more active hair cells exhibit greater susceptibility to neomycin. (**A**) Cell counts of the remaining hair cells per neuromast following neomycin treatment. Control Hoechst-: 3.63 ± 1.58; 50 µM Neo Hoechst-: 3.16 ± 1.60; 100 µM Neo Hoechst-: 2.00 ± 1.00; Control Hoechst+: 8.88 ± 1.83; 50 µM Neo Hoechst+: 2.08 ± 1.68; 100 µM Neo Hoechst+: 0.35 ± 0.65; mean cells ± SD; n = 24 neuromasts Control, 25 neuromasts 50 µM Neo, and 23 neuromasts 100 µM Neo. (**B**) Mean mitoTimer ratios for surviving hair cells following neomycin treatment. Control Hoechst-: 0.50 ± 0.09, n = 52 cells; 50 µM Neo Hoechst-: 0.45 ± 0.14, n = 64 cells; 100 µM Neo Hoechst-: 0.42 ± 0.12, n = 46 cells; Control Hoechst+: 1.03 ± 0.25 n = 124 cells; 50 µM Neo Hoechst+: 0.78 ± 0.23, n = 38 cells; 100 µM Neo Hoechst+: 0.76 ± 0.25, n = 8 cells; mean ratio ±SD; nine fish per group. Significance analyzed by Kruskal-Wallis test with Dunn's post-test.

DOI: https://doi.org/10.7554/eLife.38062.020

Constant sensory input and $Ca^{2+}$ flux likely place high demands on mitochondrial activity with potentially long-term consequences. We were able to probe the relationship between mitochondrial history and hair cell activity by monitoring mitoTimer signal in hair cells. mitoTimer fluorescence ratio (red:green) increased as hair cells functionally matured, and after MET was increased by orbital shaking (*Figures 3*, *4* and *7*). When MET activity is significantly reduced or abolished, as with benzamil treatment or in *sputnik* mutants, we observed a dramatic decrease in mitoTimer ratio (*Figure 5*). mitoTimer is a unique but complex indicator, likely reflecting multiple facets of mitochondrial biology. Thus, although there is a significant dependence on hair cell activity, an exact interpretation of mitoTimer fluorescence shifts is less clear. Based on previous studies, we expect that mitoTimer in hair cells reports mitochondrial age, in addition to oxidation, organelle turnover and/or turnover of mitochondrial protein, or a combination of these, as discussed below (*Ferree et al., 2013*; *Hernandez et al., 2013*; *Laker et al., 2014*). mitoTimer as a reporter of mitochondrial stress and oxidation was first described in a study by *Laker et al. (2014)*. mitoTimer red fluorescence shifts were observed in cultured myoblasts and *Drosophila* heart tube after treatment with agents known to increase mitochondrial ROS production, including rotenone, antimycin A, and paraquat. Additionally, the authors observed a significant red shift in aging flies, suggesting that mitoTimer reflects both the age and redox history of the organelles. Later studies also revealed that mitoTimer can be used to assess oxidative stress following ischemia-reperfusion injury in muscle (*Wilson et al., 2019*). These findings are consistent with studies of the formation of DsRed fluorescent protein, which in itself is mediated by oxidation (*Verkhusha et al., 2004*; *Yarbrough et al., 2001*). In lateral line hair cells, we observed shifts in the mitoTimer fluorescence ratio relative to increased or decreased MET. Given that altered MET is likely to have a substantial metabolic impact on the cells, the change in mitoTimer fluorescence could very well reflect corresponding changes in mitochondrial stress and oxidation over time. This is bolstered by our observation of (1) increased hair cell oxidation with sustained stimulation as reported by CellROX; and (2) reduced JC-1 fluorescence in the absence of MET, indicating that MET acutely influences mitochondrial polarization (*Figures 7* and *2*).

In addition to reporting mitochondrial redox history, mitoTimer has also been used to investigate mitochondrial dynamics and transport (*Ferree et al., 2013*; *Hernandez et al., 2013*). The decreased

mitoTimer red:green ratio observed in *sputnik* mutants could be interpreted to reflect increased mitochondrial import (with a corresponding increase of 'new' green protein relative to red). An important driving factor for mitochondrial import is mitochondrial membrane potential (*Dudek et al., 2013*; *Wiedemann et al., 2004*); however we did not observe a correspondence between potential and mitoTimer fluorescence ratio. We observed a decrease in mitochondrial polarity in *sputnik* mutants with JC-1 staining, which in isolation would be predicted to result in an increase in the red:green ratio of mitoTimer, rather than the decrease we observed. Decreased red: green ratio might also be interpreted as reflecting increased protein turnover (with degradation of "old" red protein) or increased organelle turnover caused by changes in processes such as mitophagy. We sought to further examine this idea in the context of MET mutants by monitoring changes in red mitoEos fluorescence after photoconversion. A similar strategy has been used previously in mice using the photoconvertible protein Dendra2 (*Pham et al., 2012*). Although this analysis provides only an estimate of turnover (based on the loss of photoconverted protein), our results suggest that increased mitochondrial turnover or turnover of protein occurs when MET activity is reduced (*Figure 6*), consistent with the observed changes in mitoTimer. This difference in mitochondrial protein turnover in *sputnik* mutants was unexpected. Previous studies have demonstrated mitochondrial turnover via mitophagy occurs as a quality control process (*Campanella et al., 2009*; *Twig et al., 2008*; *Nowikovsky et al., 2007*; *Kim et al., 2007*; *Mizushima et al., 2003*; *Zhao et al., 2002*). With increased mitochondrial activity in MET active hair cells, we might predict that these cells would incur more damage over time and, as a result, turnover would be higher compared to *sputnik* mutant siblings. One possibility is that mitochondria in wildtype hair cells undergo more fusion and elongation relative to *sputnik* mutants. In other cell types, increased OXPHOS activity has been associated with elongated mitochondrial morphology and, in studies of isolated mitochondria, OXPHOS has been shown to stimulate fusion (reviewed *Mishra and Chan (2016)Mishra and Chan, 2016*). The fact that mitochondrial activity and dynamics have been shown to reciprocally regulate each other suggests that mitoTimer could also reflect the interrelatedness of the two, as well as mitochondrial turnover via mitophagy. Reduced mitophagy has been observed with increased ROS in neurons (*Qi et al., 2011*). If similarly true in hair cells, increased or sustained MET activity would predictably lead to red-shifted mitoTimer fluorescence. Studies examining mitochondrial dynamics relative to MET will be important for further elucidating this relationship.

The irreversible nature of mitoTimer conversion afforded us the benefit of examining the cumulative effect on mitochondria over time, but it does not provide a dynamic picture of mitochondrial activity. It should also be noted that our manipulation of hair cell activity is limited to MET, although other aspects of hair cell function contribute to energy consumption, for example neurotransmitter packaging and release. A recent report demonstrates that there is heterogeneity amongst hair cells in their synaptic response to mechanical stimulus (*Zhang et al., 2018*). Future studies could assess the contribution of basal cellular activity to mitochondrial metabolism and dynamics, as well as a correspondence with hair cell age or susceptibility.

## Selective susceptibility of lateral line hair cells

Early studies of lateral line hair cell susceptibility to aminoglycoside antibiotics demonstrated that hair cell loss is reliably dose-dependent (*Harris et al., 2003*; *Santos et al., 2006*), where lower doses of neomycin lead to a smaller percentage of hair cell loss. Upon entering hair cells, neomycin disrupts $Ca^{2+}$ handling and causes increased mitochondrial ROS production prior to hair cell death (*Esterberg et al., 2014*; *Esterberg et al., 2016*). Blocking mitochondrial $Ca^{2+}$ uptake as well as treatment with ROS scavengers that specifically target mitochondria mitigate neomycin-induced damage, highlighting the importance of mitochondrial distress in hair cell susceptibility. The fact that some lateral line hair cells die and others survive has been perplexing. Our data suggest that baseline $Ca^{2+}$ levels or relative levels of mitochondrial membrane polarization just prior to neomycin exposure do not correspond with hair cell susceptibility. Rather, the cumulative impact of mitochondrial activity over a longer period of time serves as better indicator of susceptibility.

Given the influence of MET on mitochondrial activity and the vulnerability of mitochondria to neomycin exposure, we hypothesize that the accumulation of mitochondrial stress underlies differential hair cell susceptibility. Live imaging studies with mitoTimer reveal that cells with a red-shifted fluorescence ratio are more susceptible to low-dose neomycin exposure (*Figure 9*). As older cells, they have been mechanotransducing for a longer period of time (*Figure 4*), but they are not necessarily

more active than any other cell, or the most active cells at the time of exposure. We previously showed that maturation of individual hair cells corresponds with neomycin sensitivity: with hair cells in 4dpf fish relatively insensitive to neomycin exposure compared to 5dpf fish (*Santos et al., 2006*). Here we build on this finding to show an age-dependent difference in cumulative mitochondrial activity that correlates with neomycin susceptibility (*Figure 10*). Hair cells are active sensory receptors receiving constant sensory input. With this in mind, we propose that the normal activity sustained over the life of a hair cell contributes to cumulative mitochondrial insult. As a result, older cells are less capable of overcoming acute mitochondrial insult and are more likely to succumb to neomycin-induced damage. Alternatively, hair cells that have more active mitochondrial turnover or repair may be less susceptible to damage. Experiments looking at turnover in mitochondrially-targeted photoconvertible Eos protein (*Figure 6*) suggests that MET activity is associated with decreased mitochondrial turnover. Thus, MET activity might influence hair cell vulnerability not by cumulative mitochondrial damage but by altering mitochondrial turnover and repair. In this scenario, cells with lower mitoTimer red:green ratios may be less susceptible to neomycin exposure because they have more active mitochondrial repair or regeneration.

We note that our experiments assessing mitoTimer and susceptibility to neomycin exposure are correlative. We find that mitoTimer red:green ratios increase with age and with activity, and correspond to increased susceptibility to damage. However other age-related (or activity-related) factors may play a role in determining the relative vulnerability of hair cells to damage either in addition to mitochondrial changes or independently. Although we observe a decrease the relative red:green ratio of mitoTimer by blocking MET activity, reductions in MET also protect hair cells by blocking uptake of neomycin (*Hailey et al., 2017*), preventing independent evaluation of changes in mitoTimer and susceptibility. We do find significant shifts in mitoTimer fluorescence after placing fish on an orbital shaker, allowing a test of whether changes in mitoTimer ratio caused by this procedure alter neomycin susceptibility. We found that a 24-hour incubation on the orbital shaker had no effect on dose dependent hair cell loss after neomycin exposure compared to controls (data not shown). These results demonstrate that mitochondrial changes induced by orbital shaking are not sufficient to alter susceptibility. It may be that the relative mitochondrial changes are just not substantial enough under these conditions to effect changes in neomycin susceptibility, or that induced changes are irrelevant. We do not know whether orbital shaking might also contribute confounding factors such as MET inhibition after long term stimulation.

## Connections to mammalian hair cell susceptibility

In the cochlea, differential susceptibility and hair cell loss occurs based on hair cell type and location along the tonotopic axis, where outer hair cells are more vulnerable than inner hair cells, and high frequency hair cells more sensitive than low frequency hair cells (*Forge and Schacht, 2000*; *Kopke et al., 1999*). Our findings are consistent with studies that implicate disparities in metabolic and oxidative stress between cochlear hair cells as an underlying factor of differential susceptibility (*Jensen-Smith et al., 2012*; *Sha et al., 2001*). It is also worth noting that hair cell vulnerability to ROS may also increase with age, as a number of redox-regulating mechanisms display age-related changes. For example, expression of the ROS neutralizing enzyme SOD2 decreases in mammalian hair cells over time (*Jiang et al., 2007*). Expression of the mitochondrial Sirtuin enzymes, particularly SIRT3, have also been shown to decrease with increasing age in the mammalian cochlea (*Takumida et al., 2016*). Interestingly, SIRT3 activity has been shown to delay the onset of age-related hearing loss and protect hair cells from both noise and aminoglycoside-induced hair cell damage due to its role in antioxidant defense (*Someya et al., 2010*; *Brown et al., 2014*; *Quan et al., 2015*). Together, these studies suggest that the loss of ROS regulation over time contributes to hair cell susceptibility and hearing loss. This is further supported in work from *Someya et al. (2009)*, where overexpression of mitochondrially-targeted catalase—another antioxidant enzyme—attenuated hair cell loss and improved hearing thresholds in aging mice (*Someya et al., 2009*). Although it appears that mitochondrial oxidation may increase over time in lateral line hair cells, changes in the expression of redox enzymes through maturation have not been examined. Extending analysis of mitoTimer expression to the mammalian cochlea would provide valuable insight into selective susceptibility of mammalian hair cells and a useful comparison to studies focused on the lateral line.

The relationship between mitochondrial dysfunction and hearing loss is of long standing interest. Mitochondrial impairment has been identified in multiple types of cochlear damage and several mutations in mitochondrial genes, as well as nuclear-encoded mitochondrial proteins, are known to cause both syndromic and non-syndromic deafness (*Kokotas et al., 2007*; *Ding et al., 2013*; *Luo et al., 2013*). Moreover, in some cases the mitochondrial mutations appear to synergize with environmental insults, leading to more extensive cochlear damage. For instance, the A1555G mutation in the MTRNR1 gene encoding 12S rRNA has been shown to dramatically predispose patients to hearing loss after exposure to aminoglycoside antibiotics (*Prezant et al., 1993*; *Fischel-Ghodsian et al., 1997*). This supports the possibility that as hair cells experience cumulative mitochondrial stress due to normal activity, as indicated in our study, the cellular response to bouts of acute damage may become further compromised over time. This has particular relevance for age-related hearing and hair cell loss. While the exact cause remains unknown, mitochondrial stress has been proposed as a possible mechanism (*Seidman et al., 2004*). It is also important to consider the complexity of mitochondrial changes that occur over time. In aging and aging-related diseases, mitophagy and mitochondrial dynamics are altered, which can in turn influence OXPHOS and ROS production (*Sebastián et al., 2017*). Given that the life of a lateral line hair cell is about 10 days, studies of mitochondrial activity, dynamics, and mitophagy in lateral line hair cells could provide valuable insight into age-related hair cell loss and hearing impairment in humans, but with a much shorter time frame. A more thorough understanding of the stresses incurred during the normal activity of these high-energy cells will inform therapeutic efforts to protect hair cells and hearing function.

# Materials and methods

## Key resources table

| Reagent type (species) or resource | Designation | Source or reference | Identifiers | Additional information |
|---|---|---|---|---|
| Strain, strain background (*Danio rerio*) | mitoTimer | This paper | | Tg(myosin6b:mitoTimer)[w208]; Gateway cloing and Tol2-mediated transgenesis. |
| Strain, strain background (*Danio rerio*) | mitoEos | This paper | | Tg(myosin6b:mitoEos)[w207]; Gateway cloing and Tol2-mediated transgenesis |
| Strain, strain background (*Danio rerio*) | nlsEos | This paper | | Tg(atoh1a:nls-Eos)[w214]; CRISPR Knock-In |
| Strain, strain background (*Danio rerio*) | mitoGCaMP3 | PMID: 25031409 | ZFIN:ZDB-TGCONSTRCT-141008–1 | Tg(myosin6b:mitoGCaMP3)[w119] |
| Strain, strain background (*Danio rerio*) | cytoRGECO | PMID: 25114259 | ZFIN:ZDB-TGCONSTRCT-150114–2 | Tg(myosin6b:R-GECO1)[vo10Tg] |
| Strain, strain background (*Danio rerio*) | Cdh23 mutant (*sputnik*) | PMID: 9491988 | ZFIN:ZDB-GENO-170526–2 | Cdh23[tj264] mutant |
| Chemical compound, drug | Benzamil | Sigma-Aldrich | Sigma-Aldrich: B2417-50MG | |
| Chemical compound, drug | Hoechst 33258 | ThermoFisher | ThermoFisher: H3569 | |
| Software, algorithm | GraphPad Prism | GraphPad Software | www.graphpad.com | |

*Continued on next page*

| Software, algorithm | Slidebook | Intelligent Imaging Innovations (3i) | www.intelligent-imaging.com |
|---|---|---|---|
| Software, algorithm | Fiji | PMID: 22743772 | |
| Software, algorithm | MATLAB | MathWorks | www.math works.com |

## Fish

Experiments were conducted on larval zebrafish, ages 5–7 days post-fertilization (unless otherwise noted), prior to sex determination. Larvae were randomly selected for experimental groups. Larvae were raised in embryo medium (14.97 mM NaCl, 500 µM KCL, 42 µM $Na_2HPO_4$, 150 µM $KH_2PO_4$, 1 mM $CaCl_2$dehydrate, 1 mM $MgSO_4$, 0.714 mM $NaHCO_3$, pH 7.2) at 28.5°C. All wildtype animals were of the AB strain. Zebrafish experiments and husbandry followed standard protocols in accordance with University of Washington Institutional Animal Care and Use Committee guidelines.

## Transgenesis and mutant fish

Two genetically-encoded indicators were cloned using the Gateway Tol2 system (Invitrogen) to generate constructs under the hair cell-specific promoter, *myosin6b* (*Obholzer et al., 2008*), and were maintained as transgenic lines: (1) mitoTimer, a reporter of mitochondrial age, oxidative stress, and turnover; and (2) mitoEos, the photoconvertible fluorophore, Eos, targeted to the mitochondria. Tg [*myo6b:mitoTimer*][w208] line was generated using a previously described DsRed mutant, DsREd1-E5, with a cytochrome C oxidase subunit VIII localization sequence (*Hernandez et al., 2013*; *Laker et al., 2014*). The Tg[*myo6b:mitoEos*][w207] line was generated similarly, with mitochondrial matrix targeting achieved by Eos fusion with the human cytochrome C oxidase subunit VIII localization sequence.

The Tg[*myosin6b:R-GECO1*][vo10Tg] line has been previously described and was provided as gift from Katie Kindt (National Institute of Deafness and Other Communication Disorders, Bethesda, MD, USA) (*Maeda et al., 2014*). The c*adherin23*[tj264] mutants (also referred to as *sputnik*) have been previously described and were provided as a gift by Teresa Nicolson (Oregon Health and Science University, Portland, OR, USA) (*Nicolson et al., 1998*). We have shown previously that the Tg[*myosin6b:mitoGCaMP3*][w119] line allows reliable detection of mitochondrial $Ca^{2+}$ (*Esterberg et al., 2014*).

## Spinning disk confocal imaging

Imaging was performed using an inverted Marianas spinning disk system (Intelligent Imaging Innovations, 3i) with an Evolve 10 MHz EMCCD camera (Photometrics) and a Zeiss C-Apochromat 63x/1.2 numerical aperture water objective. Except where noted, all imaging experiments were conducted with larvae at 5–7 days post-fertilization immersed in embryo media containing 0.2% MESAB (MS-222; ethyl-3-aminobenzoate methanesulfonate). Fish were stabilized using a slice anchor harp (Harvard Instruments) so that neuromasts on immobilized animals had access to the surrounding media. Imaging was performed at ambient temperature, typically 25°C. Fish were oriented in two different positions for in vivo imaging studies. For waterjet experiments, fish were placed ventral side up and neuromasts OC1, D1, or D2 were imaged (*Raible and Kruse, 2000*). For all other imaging studies, fish were positioned on their sides against the cover glass to image neuromasts from the posterior lateral line.

## In vivo Waterjet Stimulation and Image Analysis

Waterjet stimulation was used to stimulate lateral line hair cells via directional displacement of the stereocilia. This consisted of a rectified sinusoidal pressure wave applied at 1 or 10 Hz (*Trapani and Nicolson, 2010*). A glass pipette was filled with extracellular solution and placed approximately 100 µm from the hair cells of a given cluster, or neuromast. Pressure output was driven by a pressure clamp (HSPC-1, ALA Scientific) and movement of the kinocilia was used to confirm pressure wave application. GCaMP and RGECO fluorescence values were acquired with a 1 s capture interval using Slidebook software (3i). For waterjet stimulation paired with time-lapse imaging, camera intensification was set to maintain exposure times at approximately 100 ms for GCaMP and RGECO, keeping

pixel intensity <25% of saturation. Camera gain was set to three to minimize photobleaching. GCaMP3 fluorescence was acquired with a 488 nm laser and 535/30 emission filter. RGECO fluorescence was acquired with a 561 nm laser and 617/73 emission filter.

For analyses of waterjet hair cell responses, fluorescence measurements were exported to MATLAB. As described previously, fluorescence intensity values for the duration of the acquisition period were normalized to the baseline value prior to stimulation (*Sebe et al., 2017*). MATLAB scripts were used to compute the integrated area of the response, peak of response, and time to reach 75% of the peak (rise time to 75%) (*Esterberg et al., 2014*; *Sebe et al., 2017*). The integrated area of the response is the sum of fluorescence intensity values captured during the waterjet stimulus. The 75% rise time is the number of seconds it takes from the start of the stimulus to reach 75% of the peak value.

## mitoTimer and JC-1 imaging and analysis

For static mitoTimer imaging studies, zebrafish were immobilized with a harp and Z-sections were taken at 2 μm through the depth of the neuromast, typically 16 μm. Camera intensification was set at 650 and gain at three in order to keep exposure times less than 500 ms and maximum intensities less than 75% of saturation. Settings were held constant across experiments that were directly compared. Green mitoTimer fluorescence was acquired with a 488 nm laser and 535/30 emission filter, while red fluorescence was acquired with a 561 nm laser and 617/73 emission filter.

When mitoTimer imaging was conducted in combination with vital dye staining, live swimming zebrafish were incubated in embryo media containing Hoechst 33258 (10 mg/ml, ThermoFisher) diluted at 1:5000 for 30 min and washed 3 times in embryo media. Larvae were then immobilized for imaging as described above, 24 hr after treatment. Hoechst fluorescence was captured with a 405 nm laser and 445/45 emission filter. 2 s exposures were used to capture dim signals that persisted 24 hr after labeling. Due to the dimness of the Hoechst signal and the high auto-fluorescence of larvae in this wavelength, we considered fluorescence intensity greater than 1 SD over background to be label retaining. For experiments shown in *Figure 7B*, *Figure 4—figure supplement 1*, and *Figure 10B*, mitoTimer and Hoechst fluorescence imaging were conducted using a Zeiss LSM 880 confocal microscope and a C-Apochromat 40x/1.2 numerical aperture water objective at a 5x digital zoom. Fluorescence gain levels were kept consistent across samples for each experiment. Z-sections were taken at 2 μm through the depth of the neuromast. Fish were positioned on their sides against the cover glass to image neuromasts from the posterior lateral line.

For analyses of mitoTimer, Z-stacks of Tg[*myo6b:mitoTimer*][w208] labeled hair cells were opened in 3i SlideBook or Fiji software (*Schindelin et al., 2012*). Using the maximum projection of each image, mean intensity values were extracted by drawing masks across entire neuromasts that were exported for analysis. Fluorescence intensities were calculated less the mean background signal (based on a 30 μm x 30 μm background mask that excludes hair cells). For dye labeling studies, masks were generated for individual cells within neuromasts. For single cell analyses, the median ratio for mitoTimer red and green fluorescence was calculated for all distinguishable cells within a given neuromast. Then, each cell's ratio was normalized to the median value of its cluster.

The same imaging and analysis protocol was followed for static imaging of zebrafish labeled with JC-1 dye (ThermoFisher). To load the dye, free-swimming larvae were incubated in 1.5 μM JC-1 (diluted in embryo media) for 30 min and then washed three times in embryo media. Imaging was conducted 90 min after JC-1 incubation.

## CellROX imaging and analysis

We have shown previously that CellROX green dye allows reliable detection of hair cell oxidation (*Esterberg et al., 2016*). Free-swimming zebrafish larvae were incubated for 24 hr in 2.5 μM CellROX Green (ThermoFisher) diluted in embryo media. CellROX fluorescence was acquired with a 488 nm laser and 535/30 emission filter. Image acquisition and analysis was conducted as described above for static imaging, but with a notable difference. Because the dye labels both hair cell and support cell nuclei, image analysis was conducted in a single plane. Mean intensity values were extracted by drawing a mask around all cells that were reliably distinguished as hair cells based on nucleus shape and location. Mean intensity was then normalized by the area of the mask to control for differences in the number of distinguishable cells.

## mitoEos imaging and analysis

Tg[*myo6b:mitoEos*]^w207 larvae were exposed to UV illumination with a AUKEY LT-SET1 UV flashlight for 10 min with 1 min rest. This was repeated three times for a total of 30 min of UV exposure to maximize photoconversion of Eos protein. Imaging and analysis were conducted as described above for static imaging studies at the time points indicated.

## Time-lapse imaging

For time-lapse imaging studies, baseline fluorescence readings were taken prior to aminoglycoside exposure. Neomycin sulfate (Sigma, St. Louis, MO) was then added as a 5x concentrated stock to achieve a final concentration of 50 µM. Fluorescence images were captured in 3 min intervals for 75 min. A motorized stage with set x, y, and z coordinates allowed acquisition from multiple neuromasts per fish during time-lapse recordings. For imaging of all indicators, camera gain and intensification were set to minimize photobleaching and to keep baseline pixel intensity less than 25% of saturation. Z-sections were taken at 2 µm intervals through the depth of the neuromast, typically 16 µm. The same imaging strategy was used for all indicators, however, images were collected at 2 min intervals with mitoGCaMP3 and cytoRGECO. Time-lapse images were aligned using 3i SlideBook software. Baseline fluorescence was extracted as noted above for single cells. As described previously, cells were categorized as living or dying based on their fragmentation and clearance from the neuromast following neomycin exposure (*Esterberg et al., 2013*).

## Orbital stimulation

4dpf larval zebrafish were placed on a BioExpress S-3200-LS orbital shaker at 60 RPM for 24 or 48 hr. For the control condition, fish were placed adjacent to the orbital shaker. In both conditions, 100 × 20 mm dishes with a volume of 30 ml of embryo embryo medium were used. All experiments were conducted at room temperature.

## Assessment of hair cell turnover with nuclear-localized eos

Transgenic larvae expressing hair cell-specific membrane-targeted GFP, Tg[*Pou4f3:gap43-GFP*] (*Xiao et al., 2005*), larvae crossed to larvae expressing nuclear localized Eos under the control of the Atoh1a regulatory region, Tg[*Atoh1a:nls-Eos*]^w214. Photoconversion was performed as described for mitoEos just prior to 48 hr benzamil (200 µM) or 0.5% DMSO control treatment. At 5dpf, all hair cells were counted and categorized as red label-retaining or not.

The Tg[*Pou4f3:gap43-GFP*] line was a gift from H. Baier and has been previously described (*Xiao et al., 2005*).The Tg[Atoh1a:nls-Eos]^w214 line was generated using CRISPR knock-in methodology (*Kimura et al., 2015*). The donor DNA construct was assembled using the Gateway/ Tol2 system, introducing the mbait sequence in front of the hsp70 promoter sequence. Eos was targeted to the nucleus using the SV40 nuclear localization sequence PKKKRKV (*Curran et al., 2010*). Guide RNAs were synthesized according to the protocol outlined in *Shah et al. (2015)*. Guide RNAs were purified using a Zymo RNA Clean and Concentrator kit, diluted to 1 µg/µl, aliquoted into 4 µl aliquots, and stored at −80°C. Injection mix contained 200 ng/µl Atoh1a gRNA, 200 ng/µl mbait gRNA, 800 ng/µl Cas9 protein (PNA Bio #CP02), and 20 ng/ul mbait-hsp70-nlsEos plasmid. The gRNAs and Cas9 protein were mixed together first, then heated at 37°C for 10 min, after which the other components were added. 1–2 nl of injection mix were co-injected into single-cell AB embryos. Larvae were screened at three dpf for fluorescence, tested for photoconversion, and raised to adulthood. F0 adults were outcrossed to ABs and screened for Eos fluorescence. Transgene-positive F1s were raised and maintained as a stable line. Guide RNA target sequences: *atoh1a*: GGAGACTGAA TAAAGTTATG; Mbait: GGCTGCTGCGGTTCCAGAGGTGG (*Kimura et al., 2015*).

## Quantification and statistical analysis

For this study, a biological replicate is defined as an individual cell or an individual fish as indicated in the figure legend or text. Cells were analyzed across neuromasts and fish as reported in figure legends. Values for each fish represent the mean measurements of 2–4 neuromasts as reported in figure legends. Data for quantification and statistical comparisons are taken from single experiments (unless otherwise indicated), although each experiment was repeated at least twice to confirm results. Sample sizes were estimated during study design based on previous

experiments (*Esterberg et al., 2013*; *Esterberg et al., 2016*; *Sebe et al., 2017*). Zebrafish larvae were randomly selected and sorted into experimental groups. Masking was not used during data collection or analysis. Statistical parameters, including reporting mean and SD or SEM as well as n values, are stated both in the results section and the figure legends for each experiment described. A p value of less than 0.05 was considered significant. GraphPad Prism 6.0 software was used for all statistical analyses and graphical representations.

## Acknowledgements

The authors thank David White and the UW Fish Facility staff for animal care and Max Turner and Jonathan David Ramos for assistance with data analysis. We thank Katie Kindt for the Tg[*myo6b:R-GECO₁*] line and Teresa Nicolson for the *sputnik* mutants.

## Additional information

### Funding

| Funder | Grant reference number | Author |
| --- | --- | --- |
| National Institute on Deafness and Other Communication Disorders | R01DC015783 | David W Raible |
| National Science Foundation | DGE-1256082 | Sarah B Pickett |
| National Institute on Deafness and Other Communication Disorders | T32DC536115 | Sarah B Pickett |

The funders had no role in study design, data collection and interpretation, or the decision to submit the work for publication.

### Author contributions

Sarah B Pickett, Conceptualization, Formal analysis, Investigation, Methodology, Writing—original draft; Eric D Thomas, Investigation, Writing—review and editing; Joy Y Sebe, Conceptualization, Formal analysis, Investigation, Writing—review and editing; Tor Linbo, Methodology, Writing—review and editing; Robert Esterberg, Conceptualization, Methodology, Writing—review and editing; Dale W Hailey, Visualization, Methodology, Writing—review and editing; David W Raible, Conceptualization, Data curation, Formal analysis, Supervision, Funding acquisition, Project administration, Writing—review and editing

### Author ORCIDs

Sarah B Pickett (D) http://orcid.org/0000-0003-1657-1660
David W Raible (D) http://orcid.org/0000-0002-5342-5841

### Ethics

Animal experimentation: All of the animals were handled according to approved institutional animal care and use committee (IACUC) protocols (#2997-01) of the University of Washington.

### Decision letter and Author response

Decision letter https://doi.org/10.7554/eLife.38062.025
Author response https://doi.org/10.7554/eLife.38062.026

## Additional files

### Supplementary files

• Transparent reporting form
DOI: https://doi.org/10.7554/eLife.38062.023

**Data availability**

All data generated or analysed during this study are included in the manuscript and supporting files.

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
