## [Decision Letter]

Thank you for submitting your article "Cumulative mitochondrial activity confers selective susceptibility in active mechanosensory hair cells" for consideration by *eLife*. Your article has been reviewed by three peer reviewers, including Dwight E Bergles as the Reviewing Editor and Reviewer #1, and the evaluation has been overseen by Didier Stainier as the Senior Editor. The following individuals involved in review of your submission have agreed to reveal their identity: Katie Kindt (Reviewer #2).

The reviewers have discussed the reviews with one another and the Reviewing Editor has drafted this decision to help you prepare a revised submission.

Summary:

This study exploits the unique accessibility of lateral line hair cells in zebrafish to investigate the relationship between hair cell activity, mitochondrial calcium, mitochondrial ROS generation and sensitivity to aminoglycoside antibiotics. The authors use a variety of fluorescent reporters – some dyes and some genetically encoded – to monitor cytosolic and mitochondrial changes in vivo, a major strength of the analysis. They define the temporal relationship between cytosolic and mitochondrial calcium levels in response to different levels of activity, showing that mitochondrial calcium lags the cytosol changes, as expected, and remains elevated much longer. More activity resulted in higher transients and were largely absent in mutants lacking functional MET channels. To assess the relationship between activity and cumulative mitochondrial ROS production, the authors use a genetically encoded probe, mitoTimer, which changes fluorescence from green to red in an age and ROS dependent manner. Although not a pure ROS sensor, the authors preform some clever experiments to explore this feature, by showing that the transition is age-dependent, requires functional MET channels and is enhanced by activity (Figure 6). Using fluorescently tagged mitochondria, the authors show that mitochondrial turnover is faster in zebrafish without functional MET channels, a somewhat unexpected finding, given the prior evidence of activity dependent ROS production. Finally, they perform what is perhaps the most significant analysis, to determine the relationship between acute and cumulative mitochondrial activity and cell death in response to neomycin, which has previously been shown to cause hair cell toxicity by manipulating mitochondrial ROS. Here they show that the immediate state of the mitochondria (calcium level) is not correlated with sensitivity, but that the cumulative mitochondrial activity is, suggesting that mitochondria accumulate damage through prolonged ROS generation, which renders them more sensitive to aminoglycoside antibiotics. The studies have been performed with care, the illustrations and text are clear, and the conclusions well supported by the experiments.

Essential revisions:

1) The section on mitochondria calcium responses is a bit cumbersome with the two different stimuli. The different response patterns for the two stimuli are confusing to follow and perhaps difficult to interpret. The authors suggest that the 10 Hz stimulus may be stronger than the 1 Hz stimulus (stimulating in two directions to inhibit and excite the hair cells) and that is why the patterns are different. Alternatively, it is possible that due to the delayed rise in mitochondrial calcium signals upon stimulation (subsection “Mitochondria respond to acute hair cell stimulation”), that a stimulatory 0.5 s deflection is not quite enough time to see a rise in signal before a relatively long inhibitory 0.5 s deflection in the other direction is given. The 10 Hz stimulus with its short stim/inhib phases of 50 ms on/off may allow for easier detection of mitochondrial signals because there is no long inhibitory stimulus. This could be tested with a 20 s pulse of 0.5 s or 50 ms stimuli in just one direction. Alternatively, the authors could focus on the data from the 10 Hz stimulus.

2) The JC-1 dye experiments need additional explanation/clarification. Subsection “Mitochondria respond to acute hair cell stimulation”: How the JC-1 dye works is not explained clearly. It is confusing what "concentrations" mean with regards to the dye, and the mitochondrial membrane potential and relative state of de- or hyper-polarization. Also, JC-1 has been shown to accumulate in energized mitochondria (increase in the red/green ratio). Perhaps stating something along these lines is more straightforward. Mitochondrial polarization is vague. Use hyper- or de-polarization to specify the change in potential. Subsection “Mechanotransduction has long-term effects on mitochondria”: It is unclear how the acute MET dependent increases in mitochondrial calcium (Figure 1) corresponds to the JC-1 dye measurements. Unless the authors can explain how the acute calcium responses would change the JC-1 ratio, these two measurements are quite different and are independent measures of mitochondria activity. The JC-1 measurements are static baseline measurements of mitochondrial membrane potential.

3) The authors present no evidence that baseline calcium levels in either the mitochondria or cytosol provide an estimate of hair cell activity or reflect something similar to the acute activity in Figure 1. Baseline measurements may not even reflect baseline calcium levels in some of cells. For example, the intensity of non-ratiometric indicators can reflect the expression level of the indicator, which could vary between cells depending on cell age and turnover of the indicator.

Overall, these type of baseline measurements are not a meaningful way to assess the MET heterogeneity within a population, but rather just before/after comparisons within an individual cell or population. If the authors could link MET activity to baseline calcium levels, this would be more meaningful. Alternatively, the authors could use the JC-1 ratio to link mitochondrial membrane potential to aminoglycoside susceptibility.

4) It is critical that the authors provide a more definitive link between MET activity, mitochondrial activity, and neomycin susceptibility. The use of Hoechst to compare ages of hair cells is quite nice, but unfortunately was not incorporated in more experiments. Overall the data show a nice correlation between hair cell age and mitochondrial activity, however a similar link between MET activation and mitochondria activity is more tenuous. Increased hair cell turnover in the *sputnik* mutants could be an alternate explanation for several of the results, and the data in Figure 6 suggest that increased activity has a minimal effect on mitoTimer ratios. It would be very helpful to see if additional time in the shaker leads to continued increases in mitoTimer red:green ratios. Finally, the crux of the manuscript is really in Figure 9 which shows increased mitoTimer red:green ratios very nicely correlate with susceptibility to neomycin. But considering some of the other concerns regarding the roles of age versus activity, this experiment really just shows that there is a likely correlation between cell age, mitoTimer red:green ratio and susceptibility. It is difficult to determine whether MET activity really plays any role. A nice experiment that might look more directly at this would be to incorporate the Hoechst aging trace with shaking activity. In that way, one could compare cells of comparable ages in fish that have either been shaken or not. This would allow looking at the correlation between MET activity, mitoTimer red:green ration and neomycin susceptibility. The authors should also consider testing whether animals on the orbital shaker (with more MET and mitochondrial activity and increases mitoTimer ratio) are more susceptible than controls (no shaker).

[Editors' note: further revisions were requested prior to acceptance, as described below.]

Thank you for resubmitting your work entitled "Cumulative mitochondrial activity confers selective susceptibility in zebrafish mechanosensory hair cells" for further consideration at *eLife*. Your revised article has been favorably evaluated by Didier Stainier (Senior Editor) and a Reviewing Editor.

The manuscript has been substantially improved but there are some remaining issues that need to be addressed before acceptance, as outlined below:

It seems that one of the main goals of the study – to link MET activity to increased susceptibility to neomycin – has not been demonstrated. The authors suggest that this is due to variability among hair cells, but they have a way of marking their age and degree of stress, so I'm not sure why they didn't subclassify the cells in this experiment. Circumstantially, the effect is predicted – mitoTimer ratio increases in older hair cells with shaking and mitoTimer ratio is correlated with increased susceptibility to neomycin. Thus, it isn't clear why the authors didn't use the age marking strategy in the shaking experiment. It seems like age has the biggest effect, but they continue to imply that age equals more MET activity, but obviously there could be more complications due to age. Also, perplexing is that using photoconversion of Eos they show that mitochondrial/mitochondria protein turnover is enhanced when MET channels are inhibited. I don't think they have adequately addressed how this issue could affect interpretation of the results, though the fifth paragraph of subsection “Mitochondrial metabolism and dynamics during normal cellular activity” highlights some potential confounds. It would be helpful if the authors could discuss how this phenomonon could affect interpretation of the finding that surviving hair cells had apparently lower mitoTimer ratios (lower mitoTimer ratio could indicate less stressed mitochondria or younger/more actively restored mitochondria). Thus, while they have provided some additional clarity, the issue of MET activity, mitochondrial state, and neomycin sensitivity is still somewhat uncertain. The title should be revised to indicate what susceptibility they are referring to.

---

## [Author Response]

Essential revisions:1) The section on mitochondria calcium responses is a bit cumbersome with the two different stimuli. The different response patterns for the two stimuli are confusing to follow and perhaps difficult to interpret. The authors suggest that the 10 Hz stimulus may be stronger than the 1 Hz stimulus (stimulating in two directions to inhibit and excite the hair cells) and that is why the patterns are different. Alternatively, it is possible that due to the delayed rise in mitochondrial calcium signals upon stimulation (subsection “Mitochondria respond to acute hair cell stimulation”), that a stimulatory 0.5 s deflection is not quite enough time to see a rise in signal before a relatively long inhibitory 0.5 s deflection in the other direction is given. The 10 Hz stimulus with its short stim/inhib phases of 50 ms on/off may allow for easier detection of mitochondrial signals because there is no long inhibitory stimulus. This could be tested with a 20 s pulse of 0.5 s or 50 ms stimuli in just one direction. Alternatively, the authors could focus on the data from the 10 Hz stimulus.

Thank you for the thoughtful remarks. We agree that the section on cytoplasmic mitochondrial calcium responses to waterjet stimuli was cumbersome. While we are intrigued by the differential mitochondrial calcium responses to 1 vs. 10 Hz stimuli, we believe that these data require a rigorous follow-up in a separate study. Given that the differential responses do not contribute to the main point of the current manuscript, we have removed the 1 Hz data from the figure and the text of the manuscript, as suggested.

2) The JC-1 dye experiments need additional explanation/clarification. Subsection “Mitochondria respond to acute hair cell stimulation”: How the JC-1 dye works is not explained clearly. It is confusing what "concentrations" mean with regards to the dye, and the mitochondrial membrane potential and relative state of de- or hyper-polarization. Also, JC-1 has been shown to accumulate in energized mitochondria (increase in the red/green ratio). Perhaps stating something along these lines is more straightforward. Mitochondrial polarization is vague. Use hyper- or de-polarization to specify the change in potential. Subsection “Mechanotransduction has long-term effects on mitochondria”: It is unclear how the acute MET dependent increases in mitochondrial calcium (Figure 1) corresponds to the JC-1 dye measurements. Unless the authors can explain how the acute calcium responses would change the JC-1 ratio, these two measurements are quite different and are independent measures of mitochondria activity. The JC-1 measurements are static baseline measurements of mitochondrial membrane potential.

We have re-written the text to clarify the explanation of JC-1 and to specify changes in mitochondrial potential. Additionally, we agree that measurement of calcium responses and JC-1 ratio are very different and were not meant to be directly related. These data were presented together in differentiating them from mitoTimer, which shifts gradually over time. We have placed discussion of the JC-1 data into a new subsection “Mitochondrial activity in the absence of mechanotransduction”.

3) The authors present no evidence that baseline calcium levels in either the mitochondria or cytosol provide an estimate of hair cell activity or reflect something similar to the acute activity in Figure 1. Baseline measurements may not even reflect baseline calcium levels in some of cells. For example, the intensity of non-ratiometric indicators can reflect the expression level of the indicator, which could vary between cells depending on cell age and turnover of the indicator.Overall, these type of baseline measurements are not a meaningful way to assess the MET heterogeneity within a population, but rather just before/after comparisons within an individual cell or population. If the authors could link MET activity to baseline calcium levels, this would be more meaningful. Alternatively, the authors could use the JC-1 ratio to link mitochondrial membrane potential to aminoglycoside susceptibility.

We agree that baseline calcium levels are complicated by the fact that they may reflect reporter expression levels as opposed to heterogeneity of MET amongst cells in the neuromast. In this case, the JC-1 ratio measurement is a more meaningful indication of mitochondrial activity relative to susceptibility. As such, the calcium measurements have been removed from Figure 8.

However, baseline fluorescence measurements have been used in other studies to compare living and dying hair cells in response to neomycin treatment (Esterberg et al., 2013) as well as synaptically active and silent hair cells within neuromasts (Zhang et al., 2018). Although they may not explicitly reflect heterogeneity of MET, we believe it is still worth acknowledging that there is no correspondence between baseline calcium indicator fluorescence measurements and hair cell susceptibility. We have removed language explicitly tying these results to calcium levels and MET activity, but we have included the data in a supplemental figure (Figure 8—figure supplement 1).

4) It is critical that the authors provide a more definitive link between MET activity, mitochondrial activity, and neomycin susceptibility. The use of Hoechst to compare ages of hair cells is quite nice, but unfortunately was not incorporated in more experiments. Overall the data show a nice correlation between hair cell age and mitochondrial activity, however a similar link between MET activation and mitochondria activity is more tenuous. Increased hair cell turnover in the sputnik mutants could be an alternate explanation for several of the results, and the data in Figure 6 suggest that increased activity has a minimal effect on mitoTimer ratios. It would be very helpful to see if additional time in the shaker leads to continued increases in mitoTimer red:green ratios. Finally, the crux of the manuscript is really in Figure 9 which shows increased mitoTimer red:green ratios very nicely correlate with susceptibility to neomycin. But considering some of the other concerns regarding the roles of age versus activity, this experiment really just shows that there is a likely correlation between cell age, mitoTimer red:green ratio and susceptibility. It is difficult to determine whether MET activity really plays any role. A nice experiment that might look more directly at this would be to incorporate the Hoechst aging trace with shaking activity. In that way, one could compare cells of comparable ages in fish that have either been shaken or not. This would allow looking at the correlation between MET activity, mitoTimer red:green ration and neomycin susceptibility. The authors should also consider testing whether animals on the orbital shaker (with more MET and mitochondrial activity and increases mitoTimer ratio) are more susceptible than controls (no shaker).

The possibility of hair cell turnover in the absence of MET – as in the *sputnik* mutants – is certainly an important point. We were unable to track hair cells with the Hoechst nuclear label in this case because we found that the *sputnik* mutants do not take up the Hoechst dye (see Figure 4—figure supplement 1). This is consistent with findings that differential uptake of other dyes is MET dependent. However, we did address this point with a transgenic line expressing nuclear localized Eos and blocking MET with benzamil treatment (see Figure 5—figure supplement 1). By photoconverting and tracking the cells in control or benzamil treated conditions, we observed no differences in hair cell turnover in either group. This suggests that decreases in the mitoTimer ratio likely reflect differences in MET rather than significant differences in hair cell age or turnover.

Per the thoughtful suggestion of the reviewers, we also conducted an experiment to combine cell age tracing and orbital shaking (Figure 7B). In this experiment, mitoTimer-expressing hair cells were labelled with Hoechst dye prior to 24 hours in control or orbital shaking conditions. Hair cells were categorized as Hoechst positive or negative, and mitoTimer fluorescence was measured. We observed an increase in the mitoTimer fluorescence ratio only for Hoechst positive cells in the orbital shaking condition. As suggested by the reviewers, we also tested whether a longer period of orbital shaking (48hrs total) lead to an increase in mitoTimer ratio and found this to be the case (Figure 7—figure supplement 2). These findings provide additional evidence of a connection between MET activity and mitochondrial activity.

In considering the reviewers suggestions, we conducted an additional experiment to further examine the correlation between MET activity, mitoTimer ratio, and neomycin susceptibility, taking into account age by using differential Hoechst labeling. For this experiment, mitoTimer-expressing hair cells were labeled with Hoechst at 4dpf. 24 hours later, the fish were treated with 50 or 100 μM neomycin. The surviving hair cells were categorized as Hoechst positive or negative, and mitoTimer fluorescence was measured (Figure 10). We observed that for both older and younger hair cells (Hoechst positive and negative, respectively), the red:green ratio of surviving cells post-neo treatment was significantly lower than cells in the untreated condition. This indicates that cells with a higher mitoTimer ratio in both older and younger populations are more susceptible to damage. Overall, these data are consistent with the idea that historically more active hair cells are more susceptible to damage.

Finally, we attempted to test whether orbital shaking treatment affected neomycin susceptibility. We placed animals on the orbital shaker for 24 hours, and then tested against increasing doses of neomycin. We found no statistical differences between groups. This finding is perhaps not surprising, given that the changes in mitoTimer we observe with shaking across the whole population are in themselves modest. We report these negative data in the Discussion section.

[Editors' note: further revisions were requested prior to acceptance, as described below.]

The manuscript has been substantially improved but there are some remaining issues that need to be addressed before acceptance, as outlined below:It seems that one of the main goals of the study – to link MET activity to increased susceptibility to neomycin – has not been demonstrated. The authors suggest that this is due to variability among hair cells, but they have a way of marking their age and degree of stress, so I'm not sure why they didn't subclassify the cells in this experiment. Circumstantially, the effect is predicted – mitoTimer ratio increases in older hair cells with shaking and mitoTimer ratio is correlated with increased susceptibility to neomycin. Thus, it isn't clear why the authors didn't use the age marking strategy in the shaking experiment. It seems like age has the biggest effect, but they continue to imply that age equals more MET activity, but obviously there could be more complications due to age.

We concur that our observations are correlative. We note that linking MET activity to increased susceptibility was not really a goal of our study. Rather we explored the relationships between MET and mitochondrial activity, age and mitochondrial activity, and mitochondrial activity and neomycin susceptibility. We do not intend to equate hair cell age with more hair cell activity, but rather note that older cells have been MET active for a longer period of time, so that MET could be a component of aging. We understand that we’ve shown relationships from A to B and B to C, but not A to C. However, we think the results are still interesting and important without this explicit connection. We have added text to more fully address the caveats about this correlative relationship to address this reviewer point (subsection “Selective susceptibility of lateral line hair cells”).

Also, perplexing is that using photoconversion of Eos they show that mitochondrial/mitochondria protein turnover is enhanced when MET channels are inhibited. I don't think they have adequately addressed how this issue could affect interpretation of the results, though the fifth paragraph of subsection “Mitochondrial metabolism and dynamics during normal cellular activity” highlights some potential confounds. It would be helpful if the authors could discuss how this phenomonon could affect interpretation of the finding that surviving hair cells had apparently lower mitoTimer ratios (lower mitoTimer ratio could indicate less stressed mitochondria or younger/more actively restored mitochondria).

We agree that this is an interesting and unexpected finding. We have added additional text to address how this finding affects interpretation of the results (subsection “Selective susceptibility of lateral line hair cells”).

Thus, while they have provided some additional clarity, the issue of MET activity, mitochondrial state, and neomycin sensitivity is still somewhat uncertain. The title should be revised to indicate what susceptibility they are referring to.

We have changed the Title to clarify this point and have softened the conclusion, reflecting the discussion above.